# Injury-activated glial cells promote wound healing of the adult skin in mice

Vadims Parfejevs[1,2], Julien Debbache[1], Olga Shakhova[3], Simon M. Schaefer[1], Mareen Glausch[1], Michael Wegner[4], Ueli Suter[5], Una Riekstina[2], Sabine Werner[5] & Lukas Sommer[1]

Cutaneous wound healing is a complex process that aims to re-establish the original structure of the skin and its functions. Among other disorders, peripheral neuropathies are known to severely impair wound healing capabilities of the skin, revealing the importance of skin innervation for proper repair. Here, we report that peripheral glia are crucially involved in this process. Using a mouse model of wound healing, combined with in vivo fate mapping, we show that injury activates peripheral glia by promoting de-differentiation, cell-cycle re-entry and dissemination of the cells into the wound bed. Moreover, injury-activated glia upregulate the expression of many secreted factors previously associated with wound healing and promote myofibroblast differentiation by paracrine modulation of TGF-β signalling. Accordingly, depletion of these cells impairs epithelial proliferation and wound closure through contraction, while their expansion promotes myofibroblast formation. Thus, injury-activated glia and/or their secretome might have therapeutic potential in human wound healing disorders.

---

[1] Institute of Anatomy, University of Zürich, 8057 Zürich, Switzerland. [2] Faculty of Medicine, University of Latvia, Raina Blvd., LV 1586 Riga, Latvia. [3] Department of Oncology, University Hospital Zürich, 8952 Schlieren, Switzerland. [4] Institute of Biochemistry, Friedrich-Alexander-Universität Erlangen-Nürnberg, 91054 Erlangen, Germany. [5] Institute of Molecular Health Sciences, ETH, 8093 Zürich, Switzerland. Correspondence and requests for materials should be addressed to L.S. (email: lukas.sommer@anatom.uzh.ch)

The skin is the largest organ of the human body and acts as the primordial barrier of the organism against the outside environment. It mainly consists of two principle components: a stratified epidermis and an underlying layer of supportive connective tissue, the dermis. In various occasions throughout life, acute injuries challenge the integrity of this frontline defence. In most cases, they trigger an immediate emergency response to establish a sealed environment and prevent blood loss and infection, but also slower, long-lasting repair mechanisms. The latter involve various cell types to restore, at least partly, the initial biological properties of the injured site[1–3]. Tissue repair mechanisms of the skin have been studied for decades and have highlighted that many key processes, such as, for instance, neo-vascularisation, are required to support the increased proliferation of fibroblasts and keratinocytes[3,4].

Besides increased blood supply, the healing response includes a second essential biological aspect: the neural response. Studies have shown that under normal circumstances hyperinnervation follows at the location of the injury[5]. Impairment of the peripheral nervous system (PNS), whether traumatic or pathologic, results in improper tissue repair and failure to heal[6]. One of the main functions of innervation has been attributed to axonal sprouting of neurons and their associated secretome of growth factors released in the wound bed upon injury[5,7,8]. However, non-neuronal cells of the PNS have also been associated with wound healing. In particular, cells expressing the progenitor marker Sox2 and originating either from nerve terminals around hair follicles (HFs), from injured peripheral nerves or from distant sites outside the regenerating dermis, were shown to be involved in skin wound healing[9]. How these cells contribute to the repair process is not entirely clear.

To specifically address the role of peripheral glia in cutaneous wound healing we used genetic mouse models allowing the tracing, conditional depletion, and conditional expansion of peripheral nerve cells in an otherwise undisturbed in vivo context.

In this study, we report a novel role of PNS glia during wound healing of the skin. After a dedifferentiation and expansion process, injury-activated glia promote wound contraction and healing. This process is mediated by the secretion of factors enhancing transforming growth factor (TGF)-β signalling, which results in increased myofibroblast formation.

## Results

**Tracing PNS glia in the injured skin.** Skin is a densely innervated organ[10] with major nerve bundles (NB) visible in both intact skin and in skin healing from full-thickness excisional wounds (Fig. 1a). To determine the potential involvement of skin innervation in wound healing, we first used genetic lineage tracing to study the fate of nerve-derived cells upon skin wounding. Tamoxifen (TM)-mediated activation of CreER$^{T2}$ in the intact skin of *Plp-CreER$^{T2}$; tdTomato* mice led to genetic tracing of peripheral glial cells in NBs of the reticular dermis, nerve terminals around HFs, as well as in nerve endings between muscle fibres[11–13]. Apart from NBs and a fraction of melanocytic cells in HFs, the epidermis and the rest of the dermis appeared void of *Plp-CreER$^{T2}$*-recombined cells (Fig. 1b, and Supplementary Figs. 1a, 2). Upon full-thickness excision wounding, however, we observed the presence of many *Plp-CreER$^{T2}$*-traced (tdTomato$^+$) single cells within the wound bed (Fig. 1c). In the intact dermis, expression of Sox10, a marker for the peripheral glial lineage[14], was mostly confined to NBs surrounded by a prominent laminin-positive perineurium. In contrast, NBs located adjacent to the wound displayed disrupted perineurial laminin expression and were associated with many single Sox10-positive (Sox10$^+$) cells that seemingly disseminated from the disrupted nerves (Fig. 1d)

and made up for the vast majority of *Plp-CreER$^{T2}$*-traced nerve-derived cells (97.4%) (Supplementary Fig. 1b).

To confirm that NBs indeed serve as a source for cells homing to the wound bed in injured mouse skin, we made use of a separate Cre line, *Dhh-Cre*, which efficiently labels the glial lineage in the PNS[15,16]. To visualise the glial lineage before and after skin injury, we employed confocal 3D-imaging techniques combined with a fructose-based tissue clearing protocol of *Dhh-Cre; tdTomato* animal samples[17]. Using this method, we observed a network of tdTomato$^+$ NBs in intact skin, while the stroma in-between the nerve fibres was devoid of labelling. In contrast, the granulation tissue of wounded skin, already 6 days post injury (D6), displayed many single cells apparently stemming from the injured nerves at the wound edge. By day 14 (D14), the entire granulation tissue was densely populated by *Dhh-Cre*-traced cells (Fig. 1e). Similar to the Sox10$^+$ cell population observed in *Plp-CreER$^{T2}$; tdTomato* mice (Fig. 1d), cells traced by *Dhh-Cre* appeared to emigrate from NBs upon injury by D6 (Fig. 1f). Interestingly, the majority of NB-derived cells found in the wound both on D7 and D14 were not associated with axons, as shown by double labelling for Sox10 and neurofilament (NF) staining (Fig. 1g, h). The combined data from separate tracing methods show that cells derived from peripheral nerves disseminate into granulation tissue upon wounding of the skin.

**Wounding induces glial de-differentiation and proliferation.** Schwann cells in peripheral nerves possess a remarkable plasticity and have been described to de-differentiate and re-differentiate following injury to the nerve[18]. To clarify the identity of nerve-associated cells populating skin wounds, we performed several immunohistological stainings for markers known to be associated with different Schwann cell states. Based on these experiments, we identified two classes of NBs located adjacent to or within the wound, which we term "quiescent" and "activated", respectively (Fig. 2a). Quiescent NBs expressed high levels of myelin basic protein (MBP), a marker of myelinating Schwann cells, and axonal NF, but showed weak expression of the low-affinity nerve growth factor receptor (p75) (Fig. 2b, d). In contrast, activated NBs were often associated with a distorted NB morphology, displayed increased levels of p75 protein expression, and stained only weakly for MBP and NF (Fig. 2c, d). These features resemble the processes of myelin and axon degradation observed in Wallerian degeneration.

In addition, only activated NBs and cells dissociating from them were positive for pERK (Fig. 2e) and c-Jun (Fig. 2g)–markers associated with Schwann cell plasticity and de-differentiation[19–21]. Importantly, cells within activated NBs displayed an increased proliferation rate (quantified as percentage of Ki67$^+$ cells) compared to quiescent nerves that were mostly composed of terminally differentiated, non-proliferative cells (Fig. 2e, f). Of note, 77% of all p75$^+$ cells also expressed c-Jun. 33% of these double-positive cells were also marked by Ki67, a nearly 10-fold increase compared to the number of p75$^+$ c-Jun$^-$ Ki67 expressing cells (3.6%) (Fig. 2g, h). These data demonstrate that wounding of murine skin results in de-differentiation and active proliferation of a peripheral glial cell population and suggest that this process allows peripheral glia to dissociate and emigrate from injured nerves.

Peripheral nerves in the skin have been shown to harbour cells that upon isolation and in culture exhibit properties of multipotent neural crest stem cells (NCSCs), from which peripheral glia develop during embryonic development[22]. To assess whether disseminating injury-activated nerve-derived cells generate NCSC lineages other than glial in vivo, we followed their fate at later time points post injury. In addition to Sox10 (which apart from

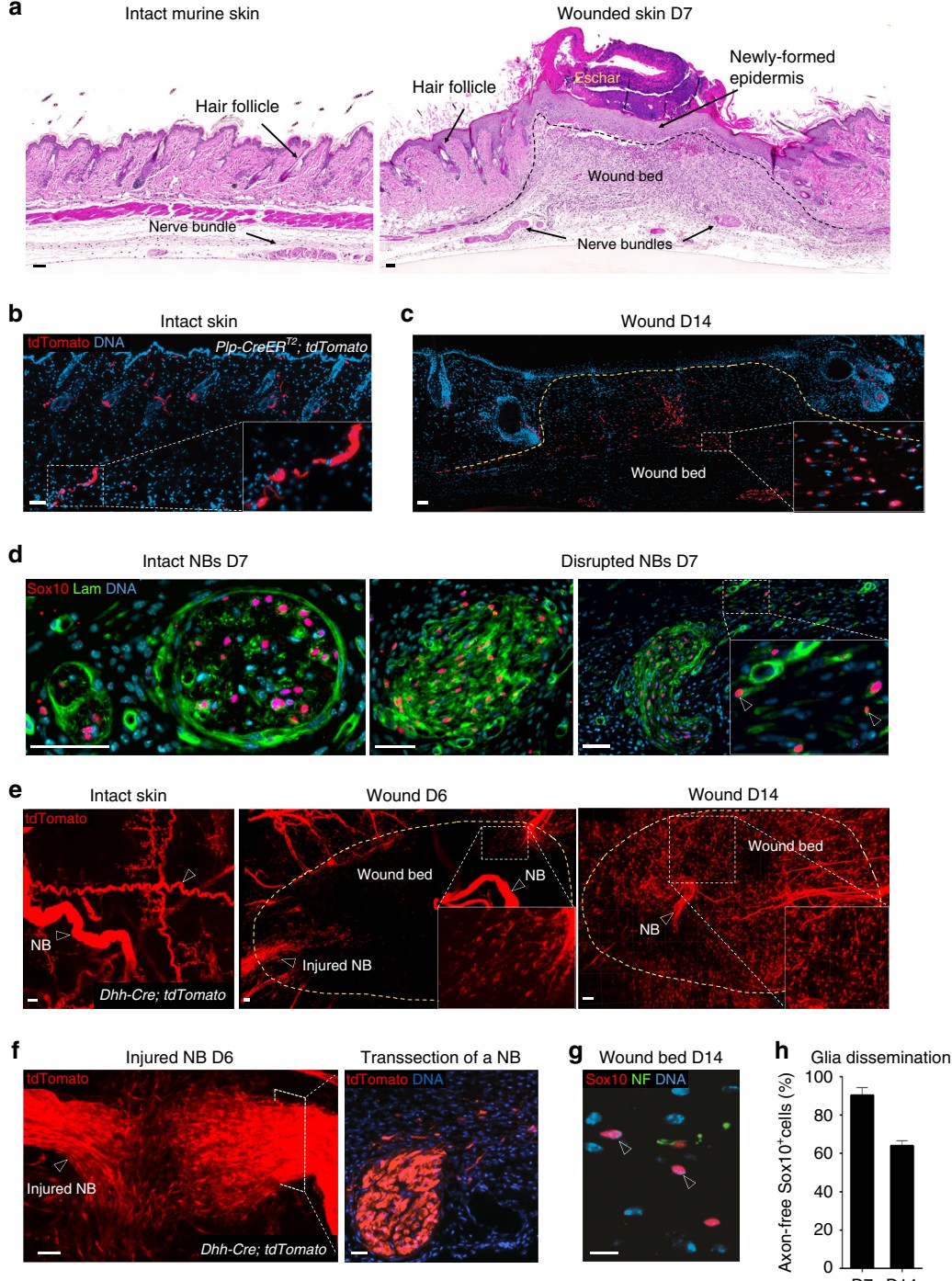

**Fig. 1** Genetic tracing of the peripheral glial lineage in intact and injured skin. **a** H&E-stained sections of intact skin and full-thickness excisional wound of adult mice at day 7 (D7) post-surgery. **b** Glial lineage tracing of *Plp-CreER^{T2}; tdTomato* intact skin of a TM-injected animal. Dermal compartment is mostly void of single-labelled cells. **c** Glial lineage tracing of *Plp-CreER^{T2}; tdTomato* injured skin at D14 post-surgery in TM-injected animal. **b**, **c** Boxed regions in the dermis are shown at higher magnification in the insets, highlighting the presence of multiple individual traced cells (red) populating the wound bed upon injury. **d** Immunofluorescence staining of skin NB for the transcription factor Sox10 and the extracellular matrix protein Laminin (Lam) in intact and D7 injured skin show disruption of perineurium and dissemination of Sox10+ cells upon injury. Arrowheads denote the presence of Sox10+/Lam+ cells outside the NB. **e** 3D imaging of the glial lineage of *Dhh-Cre; tdTomato* cleared intact mouse skin and cleared wounded skin at D6 and D14 show individual traced cells detached from NB upon injury. Arrowheads denote tdTomato-traced NBs. **f** An apparently damaged sprouting nerve was subsequently sectioned and immunolabelled for the lineage tracer tdTomato. **g** D7 and D14 wounded skin samples were subjected to immunolabelling to quantify the amount of glial cells (Sox10+) detached from NF-positive axons. Arrowheads denote glial cells with no adjacent NF staining and considered axon free. **h** The quantification of the extent of glial dissociation from axons shows that the vast majority of Sox10+ cells are not associated with axons at D7 and D14 (number of animals (*N*) = 4, number of wounds (*n*) = 8 (D7); *N* = 4 animals, *n* = 7 wounds (D14)). **b**, **c**, **d**, **f**, **g** sections were counter stained with Hoechst 33258 (blue). Scale bars, 50 μm (**a–f**), 10 μm (**g**)

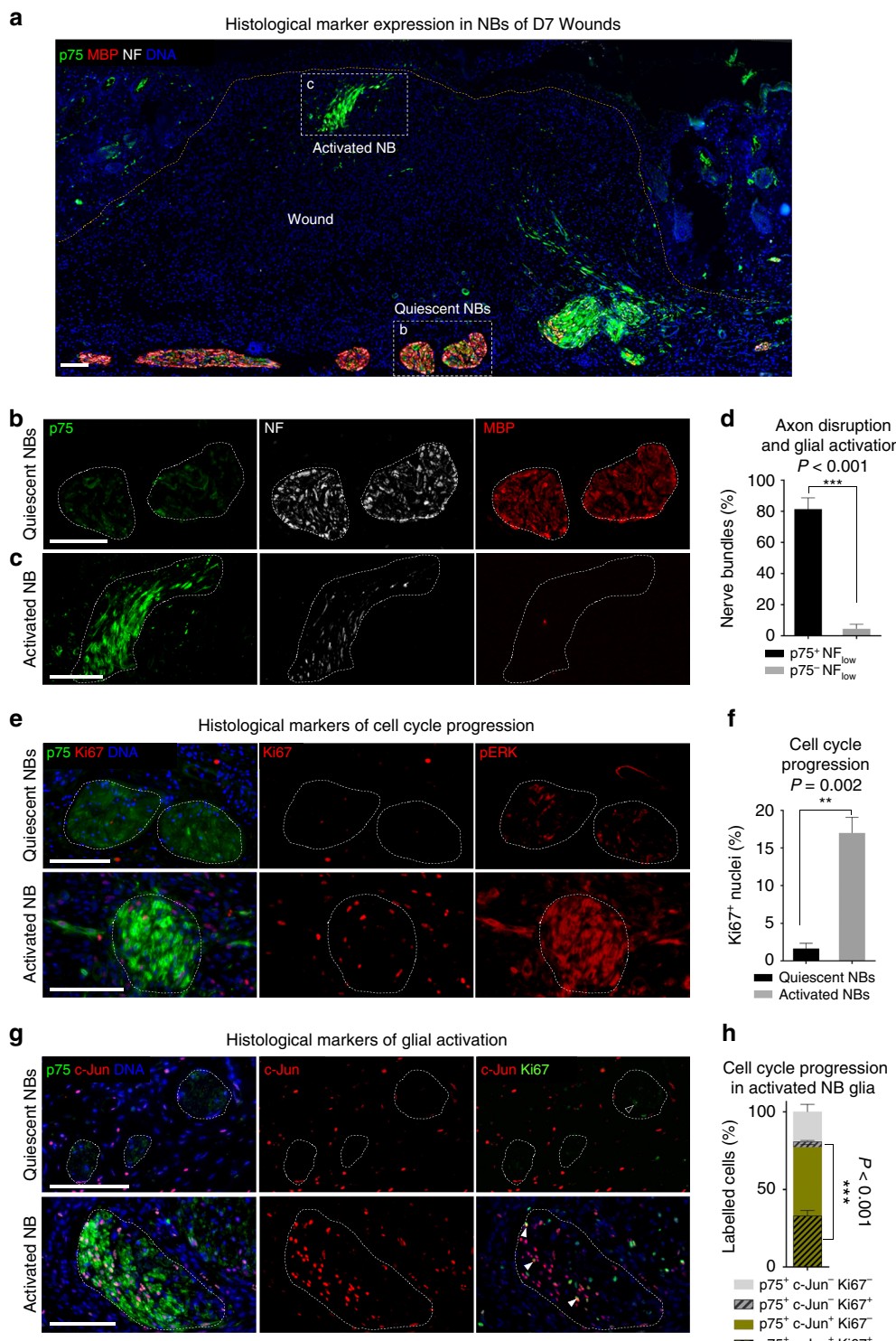

**Fig. 2** Skin injury induces peripheral glial de-differentiation and proliferation. **a** p75 (green), MBP (red), and NF (white) triple immunolabelling of wound sections shows increased expression of the de-differentiation marker p75 and loss of the differentiation marker MBP as well as loss of NF in activated NBs. **b**, **c** Single-channel insets show expression of p75, MBP, and NF in activated and quiescent NBs in injured skin. **d** Quantification of NBs categorised based on NF labelling ($NF_{High}$ and $NF_{low}$) and p75 positivity ($p75^+$ and $p75^-$). **e** Immunolabelling of pERK (marker of activated MAPK signalling pathway) and Ki67 (marker for proliferative or cycling state) (red) in quiescent vs. activated NBs in injured skin. Activated NBs are characterised by p75 staining (green). **f** Quantification of $Ki67^+$ nuclei in quiescent or activated NBs. **g** Double immunolabelling of p75 (green) and c-Jun (red) shows concomitant upregulation of both markers in activated vs. quiescent NBs in injured skin. Black arrowhead denotes a $Ki67^+$ $c-Jun^-$ cell. White arrowheads denote $Ki67^+$ $c-Jun^+$ cells. **h** Quantification of proportion of proliferative ($Ki67^+$) cells in $c-Jun^+$ and $c-Jun^-$ fractions of de-differentiated ($p75^+$) glial cells within NBs. Data are represented as mean ± SEM of $N = 7$ animals, $n = 12$ wounds for activated NBs and $N = 5$, $n = 9$ for quiescent NBs (**d**); 8 quiescent NBs from $N = 2$, $n = 4$ and 14 activated NBs from $N = 4$, $n = 6$ (**f**); $N = 6$, $n = 7$ (**h**). Scale bars, 100 μm. Wound samples are D7 post injury

glia also marks NCSCs[23]), over 90% of the disseminated cells expressed the de-differentiation marker p75 (Supplementary Fig. 1b, c). Furthermore, 77.8% of the genetically traced cells transiently lost the glial lineage marker GFAP at D7, which was however regained through D14. In addition, 77.4 ± 4% of the p75$^+$ cells displayed the activation marker c-Jun, of which 32.0 ± 3.6% were positive for the proliferation marker Ki67 (Supplementary Fig. 1d, e).

Because *Plp-CreER^{T2}* not only marks peripheral glia, but also labels a subpopulation of melanocytes within the skin (Supplementary Fig. 2a, b)[11,13], we aimed to assess the contribution of melanocytic cells to regenerating skin tissue. To do so, we compared wounding experiments in *Plp-CreER^{T2}; tdTomato* mice with *Tyr-CreER^{T2}; tdTomato* animals, in which the melanocytic lineage can be specifically traced in vivo[24,25]. Immunolabelling of wound sections from both mouse strains ruled out the presence of any Dct$^+$ cells in the granulation tissue and of any traced cells in *Tyr-CreER^{T2}; tdTomato* animals, suggesting that the melanocytic lineage does not contribute to the cell population traced by *Plp-CreER^{T2}* upon injury (Supplementary Fig. 2c, d). Although during embryonic development, neural crest cells can give rise to mesenchymal derivatives, *Plp-CreER^{T2}*-traced cells did not significantly contribute to the formation of myofibroblasts, as shown by the lack of α-smooth muscle actin (α-SMA) expression in tdTomato$^+$ cells (Supplementary Fig. 2e). Thus, activated peripheral glial cells disseminating into the wound bed appear to be lineage restricted under our experimental settings in vivo and, in particular, do not significantly contribute to the myofibroblast pool, a cell population with important function in matrix deposition and wound contraction[26].

**Activated glia secrete factors implicated in wound healing**. To address the role of injury-activated glia during wound healing, we considered the possibility that these cells provide support by paracrine secretion of factors or microenvironment modulation of the healing tissue. Such a mode of action is conceivable given that Schwann cells have been reported to upregulate chemokines as well as growth factors upon nerve injury in order to modulate inflammation and axon sprouting, respectively[27,28]. To identify glia-derived factors potentially involved in skin wound healing, we performed RNA sequencing of *Dhh-Cre*-traced nerve-derived tdTomato$^+$ cells isolated by fluorescence-activated cell sorting (FACS) from uninjured control animals and 7 days post injury (Fig. 3a).

Of the 1033 genes found to be differentially expressed between normal and wounded skin, 526 were upregulated and 506 downregulated (Fig. 3b, c). Among the genes downregulated in nerve-derived cells upon injury, we found several genes regarded as positive regulators of myelination, such as *Egr2* and *Pou3f1/Oct6*, as well as myelin genes and other genes—*Mbp*, *Mal*, and *Cdh1*—previously reported to be repressed in Schwann cells upon injury and de-differentiation[29–31]. Further, among the genes significantly upregulated upon injury were, for instance, *Ngfr*, *Gap43*, *Thy1*, *Igfbp2*, *Mdk*, *Pdgfb*, *Runx2*, *Cxcr4*, which have all been reported to be upregulated in Schwann cells upon injury in previous studies[28,29,32–34]. Consistent with previous reports, we observed similar regulation of genes implicated in axon growth, guidance and protection, including the upregulated genes *Epha5* and *Sema4f*, as well the downregulated genes *Nfasc*, *Nrn1*, and *Atp1b2*[35]. We also found a set of upregulated genes previously described as important players in Schwann cell-mediated innate immune response, chemotaxis, and myelin phagocytosis, such as *Lif*, *Cxcl5*, *Areg*, *Megf10*, *Mmp14*, which all followed a similar regulation pattern in our data set as previously described after sciatic nerve injury[21,27,28].

Because we hypothesised that injury-activated glia could exert their function through secretion of soluble factors, using MetaCore analysis from GeneGO, we focused on the predicted localisation of the gene products that were differentially expressed between non-wounded and wounded states. Intriguingly, extracellular space and membrane were the predominant localisations of gene products activated in peripheral nerve-derived cells upon wounding. Whereas most of the annotated genes from the downregulated gene list coded for proteins important in cell–cell contacts and adhesion, products of the upregulated genes tended to be secreted and associated to the extracellular matrix and extracellular space (Fig. 3d). In particular, consistent with our hypothesis of a paracrine function of injury-activated glia, the upregulated gene fraction showed a 3.5-fold enrichment in extracellular space/ECM-related genes upon injury—up to 53.4% (Fig. 3e).

When considering all differentially expressed genes coding for proteins with extracellular localisation ($n = 455$), 61.8% of them were upregulated ($n = 281$) and 38.2% downregulated ($n = 174$) (Fig. 3f). We clustered these 455 genes according to their function based on previous reports in the literature and annotations given in GeneCards (www.genecards.org). According to their described roles, we defined the following categories for upregulated genes: (1) growth factors and cell signalling, (2) TGF-β signalling, (3) chemotaxis and inflammation, (4) migration and adhesion, (5) ECM remodelling, and (6) angiogenesis. Interestingly, many genes from this list have been implicated before in different processes important for wound healing and tissue repair (Fig. 3g; Supplementary Tables 1–6).

**Depletion of injury-activated glia impairs wound healing**. In order to assess the role of injury-activated glia in an in vivo context, we decided to deplete this cell population using genetic tools. Sox10 is an important survival and maintenance factor for the Schwann cell lineage during all stages of its differentiation[23,36]. In the skin, Sox10 is exclusively expressed in the melanocytic lineage (confined to HFs in murine back skin) and the glial lineage[25,37]. To deplete nerve-derived cells in skin wounds and/or interfere with their migration into the wound, we generated *Plp-CreER^{T2}; Sox10^{lox/lox}; tdTomato* animals by breeding *Plp-CreER^{T2}; tdTomato* mice with mice carrying floxed alleles of the *Sox10* gene to specifically delete *Sox10* in the glial linage (Fig. 4a). Since effective depletion of Sox10 protein upon genetic ablation of the allele is a relatively slow process in adult mice[38], we allowed 3 weeks post-TM injection before we subjected both *Sox10* conditional knock-out (cKO) and control littermates to excisional wounding, although early enough to limit potential effects of peripheral neuropathy which peaks at 40 days post TM treatment[38]. Tissue samples were collected for analysis at several time points as depicted in Fig. 4a. NBs of *Sox10* cKO animals visually appeared similar to those of control animals and axons in quiescent NBs were preserved as shown by NF staining (Fig. 4b). To verify depletion of glial cells in the nerves, we quantified Sox10$^+$ cells within NBs located close to the wound bed and found that 56.8 ± 4.9% of all cells expressed Sox10 in control vs. 12.5 ± 3.2% in *Sox10* cKO mice at D7 (Fig. 4c).

To assess the number of injury-activated, de-differentiated glia present upon *Sox10* cKO, we performed immunofluorescence analysis and quantification of p75$^+$ and traced tdTomato$^+$ cells in the wound bed at two different time points. The data show a similar significant decrease of p75$^+$ and traced cells in *Sox10* cKO animals at both D7 (46.5 ± 4.4 vs. 7.0 ± 1.8 p75$^+$ cells mm$^{-2}$ and 40.2 ± 5.4 vs. 2.8 ± 0.9 tdTomato$^+$ cells mm$^{-2}$ in control vs. *Sox10* cKO) and D14 after wounding (90.2 ± 12.8 vs. 24.1 ± 5.0 p75$^+$ cells mm$^{-2}$ and 153.6 ± 28.8 vs. 43.9 ± 3.3 tdTomato$^+$ cells

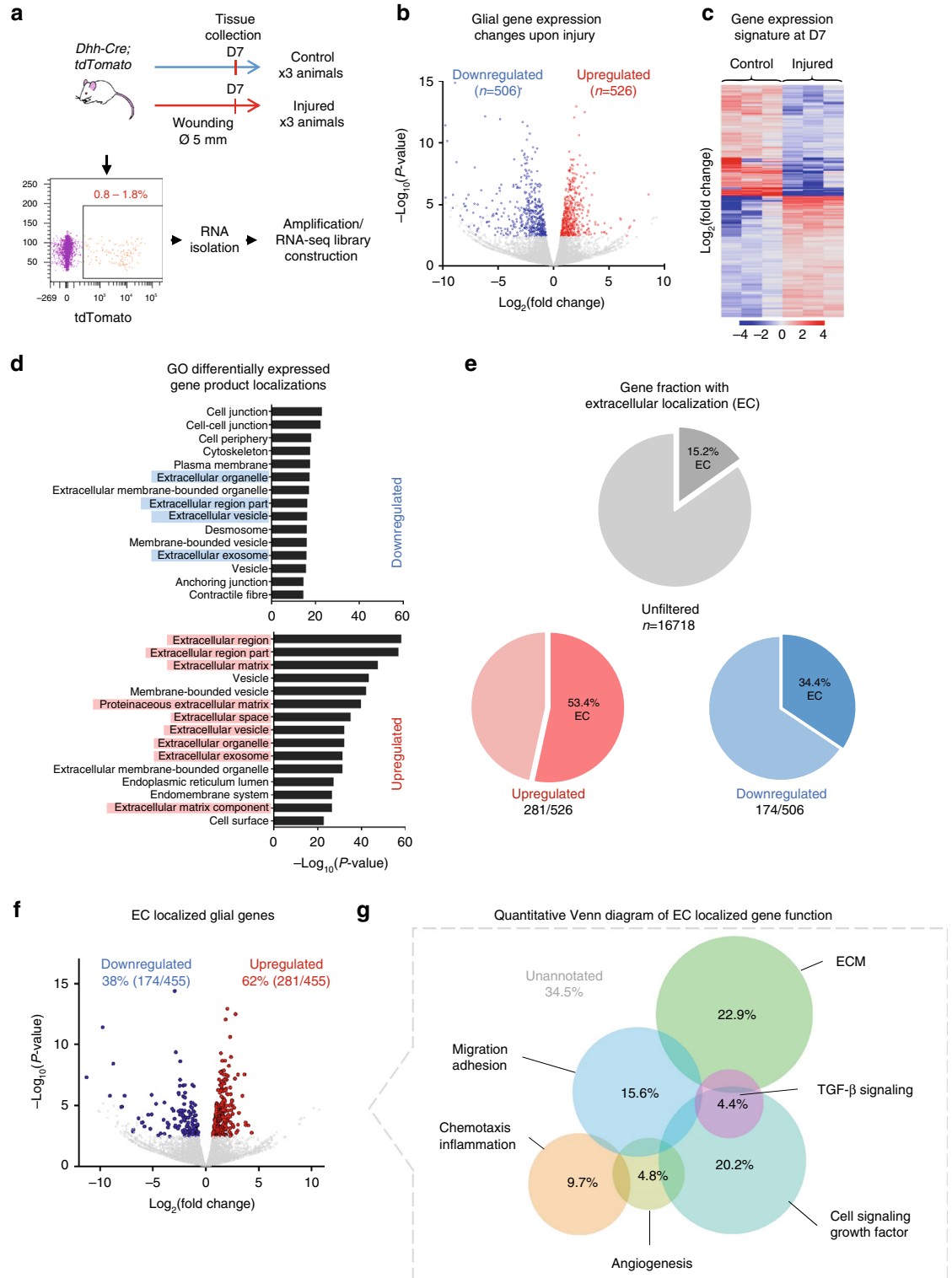

**Fig. 3** Expression profile analysis of glial lineage upon skin injury. **a** Experimental design for skin sample collection of Ctrl and D7-injured *Dhh-Cre; tdTomato* animals. Single cell suspensions were sorted for tdTomato expression using a flow cytometer (0.8–1.8% of total cell fraction). Total RNA samples of gated cells were amplified and subsequently subjected to RNA-Seq using Illumina RNA-Seq platform. **b** Volcano plot of the gene expression profile of *Dhh-Cre*-traced tdTomato[+] cells of intact skin and D7 post injury. Coloured data points meet the thresholds of FC above 1.5 and under −1.5, *P*-value < 0.05, FDR < 0.05. **c** Unsupervised sample clustering heat map of gene expression with a significantly differential expression. **d** Gene GO MetaCore prediction of the product localisation of the significantly downregulated (506 genes) and upregulated genes (526 genes) shows a significant enrichment of genes, the products of which are excreted. **e** Data representation for the percentage of gene products predicted to be destined to the extracellular space. **f** Volcano plot of the expression value of the genes with predicted extracellular localisation of their product. **g** Quantitative Venn diagram of the differentially expressed "secretome" and the associated biological function

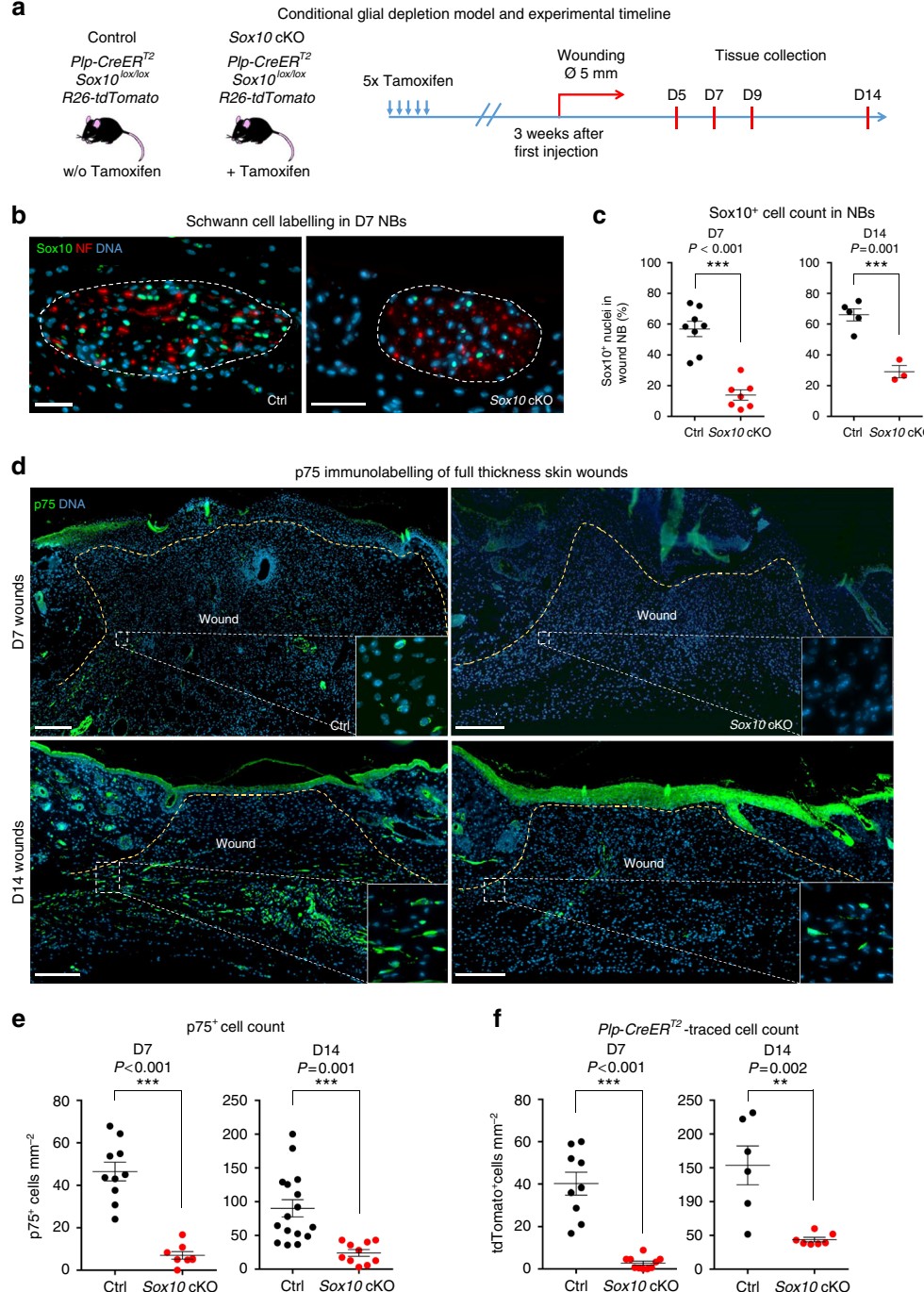

**Fig. 4** *Sox10* ablation in the glial lineage prevents invasion of injury-activated glia into the wound bed. **a** Scheme of the experimental strategy for *Plp-CreER*$^{T2}$-mediated ablation of *Sox10* in our murine wound healing model. **b** Immunolabelling of Sox10 (green) and NF (red) in skin sections shows efficient reduction of Sox10$^+$ population in NBs upon *Sox10* cKO. **c** Quantification of the percentage of Sox10$^+$ nuclei per total nuclei in NBs of injured skin of Ctrl and *Sox10* cKO animals at D7 and D14. **d** Immunolabelling of p75 (green) in skin wound sections at D7 and D14 reveals a depletion of the p75$^+$ population upon *Sox10* cKO. Boxed regions in the granulation tissue (delimited by the dotted lines) are shown at higher magnification in the insets. **e**, **f** Quantification of p75$^+$ and tdTomato$^+$ cells per mm$^2$ of granulation tissue of injured skin shows persistent reduction of both p75$^+$ and traced cells upon *Sox10* ablation at D7 and D14. Data are represented as mean ± SEM of $N = 4$, $n = 8$ (Ctrl D7), $N = 3$, $n = 5$ (Ctrl D14); $N = 4$, $n = 7$ (*Sox10* cKO D7), $N = 3$, $n = 3$ (*Sox10* cKO D14) (**c**); $N = 3$, $n = 10$ (Ctrl D7), $N = 8$, $n = 16$ (Ctrl D14); $N = 4$, $n = 8$ (*Sox10* cKO D7), $N = 5$, $n = 10$ (*Sox10* cKO D14) (**e**); $N = 4$, $n = 9$ (Ctrl D7), $N = 2$, $n = 6$ (Ctrl D14); $N = 3$, $n = 10$ (*Sox10* cKO D7), $N = 2$, $n = 7$ (*Sox10* cKO D14) (**f**). Scale bars, 50 μm (**b**), 200 μm (**d**)

mm$^{-2}$) (Fig. 4d, e, f). Further, analysis of the relatively few tdTomato-traced cells still present upon *Sox10* cKO showed that the vast majority of the fate-mapped cells, both in NBs and in the wound bed, still displayed Sox10 expression (Supplementary Fig. 3a, b). Therefore, rather than adopting alternative fates, peripheral glial cells having lost Sox10 were virtually all eliminated, in agreement with previous studies[36,38] (Supplementary Fig. 3c, d). These data demonstrate that conditional deletion

of *Sox10* in the glial linage using *Plp-CreER^T2* is an efficient method to decrease the number of peripheral nerve-derived cells in the wound.

To study possible effects caused by genetic depletion of nerve-derived cells in the wound bed, we performed macroscopic measurements and morphometric analysis of control and *Sox10* cKO wounds. Although the dynamics of wound contraction can be variable, as previously described[39], we found a significant difference in the mean wound size at D7 (14.2 ± 1.6% vs. 22.2 ± 3.1% of initial size in control vs. *Sox10* cKO) as well as at D9 (4.8 ± 1.2% vs. 11.1 ± 3.0%) (Fig. 5a, b). Since the most significant difference in wound contraction was observed at D7, we carried out morphometric analyses of D7 wound sections to assess whether other commonly measured wound healing parameters were changed as well. The principle of measurements and representative pictures of wounds at this stage are shown (Fig. 5c). Quantification of the distance of newly formed epidermis covering the wound revealed a delay in wound closure in *Sox10* cKO animals—an average of 84% of the wound was covered by epidermis in control, while only 66% of the wound was covered in *Sox10* cKO at the same time point (Fig. 5d). Likewise, the area of hyperproliferative epidermis (HPE) was significantly decreased (0.282 ± 0.017 vs. 0.220 ± 0.015 mm²) in *Sox10* cKO wounds (Fig. 5e). Importantly, in the absence of activated glia, the wound width was increased (1.99 ± 0.08 vs. 2.46 ± 0.08 mm), while the length of wound epithelium remained unchanged in *Sox10* cKO animals compared to controls, suggesting impaired contraction (Fig. 5f, g). In sum, injury-activated glia in the wound bed is required for proper healing.

**Activated glia are dispensable for angiogenesis and immune response**. Some of the genes upregulated in peripheral nerve cells upon wounding encode secreted factors, such as *Apln*, *Esm1*, *Mdk*, and *Vcan*, which have previously been shown to influence angiogenesis. Therefore, depletion of injury-activated glia could affect neovascularisation (Supplementary Tables 1 and 6). However, when compared to the granulation tissue of control wounds, genetic depletion of nerve-derived cells in the wound did neither significantly change the number nor the density of blood vessels labelled by the endothelial cell marker CD31 (Supplementary Fig. 4a).

Further, we assessed the role of these cells on immune cell chemotaxis, as our differential expression profiling revealed several genes coding for proteins known to be important for innate immune cell chemotaxis, such as *Cxcl5*, *Cxcl2*, *Lif*, and *Trem2*, among others (Supplementary Table 3). However, no detectable changes in the number of total immune cells marked by CD45 or of CD206⁺ macrophages were observed upon manipulation of the number of nerve-derived cells in the wound bed (Supplementary Fig. 4b, c).

Similarly, the above-mentioned expression profile analysis underlined significant changes in the transcription of genes involved in ECM deposition (Supplementary Table 5). Hence, we investigated the state of collagen maturation of D14 control and *Sox10* cKO wound sections using Herovici stainings. However, we could not detect obvious changes in young vs. mature collagen deposition between control and *Sox10* cKO animals with our experimental settings (Supplementary Fig. 4d).

**Glial depletion impairs TGF-β signalling and myofibroblast formation**. In the wound granulation tissue, α-SMA is a marker for myofibroblasts, which, as previously mentioned, are important for wound healing through their role in wound contraction and extracellular matrix protein (ECM) deposition[40]. Our expression profiling analysis showed upregulation of a number of

genes important for TGF-β signal regulation and myofibroblast formation, such as *Inhba*, *Ltbp2*, *Mmp2*, *Bmp1*, *Loxl2*, *Plod1*, and *Plod2* (Supplementary Table 2). Hence, we looked at the area occupied by myofibroblasts post injury. Interestingly, the area of α-SMA expression was significantly reduced in *Sox10* cKO wounds at D7 (20.7 ± 2.7% of the total granulation tissue area) compared to control (33.6 ± 2.3%) (Fig. 5h, i). To substantiate these findings, we performed western blot analysis of protein isolated from tissue of control and *Sox10* cKO wounds at D7 and measured a significant decrease of α-SMA protein levels upon *Sox10* cKO-mediated depletion of injury-activated glia (Fig. 5j and Supplementary Fig. 5a). Likewise, *Sox10* cKO wounds contained significantly reduced amounts of Periostin (Postn), a multifunctional promoter of wound healing that has been associated, among others, with formation of myofibroblasts from fibroblasts[41] (Fig. 5k and Supplementary Fig. 5b).

To investigate whether TGF-β signal activation is indeed altered in skin wounds upon ablation of the glial population, we quantified the number of cells with prominent nuclear pSMAD2 localisation—a downstream marker of activated TGF-β/activin signalling—in the granulation tissue of the wounds. At D7, depletion of activated glia led to a significant decrease in the number of pSMAD2⁺ nuclei in *Sox10* cKO vs. control wounds (11.29 ± 0.8% vs. 24.1 ± 1.8% in *Sox10* cKO vs. control, respectively) (Fig. 5l, m). In sum, conditional ablation of *Sox10* in peripheral nerves prior to injury is sufficient to prevent colonisation of the wound tissue by nerve-derived cells and leads to delayed wound contraction and re-epithelialisation.

**Activated glia promote myofibroblast formation in vivo**. Loss of the tumour suppressor gene *Pten* has previously been shown to result in proliferation of both myelinating and non-myelinating Schwann cells[42]. Hence, *Pten* inactivation might represent a genetic tool to expand the number of nerve-derived cells in wounds. Therefore, we crossed *Plp-CreER^T2* mice with mice carrying floxed alleles of the *Pten* gene. *Pten* was then depleted homozygously shortly prior to injury (Fig. 6a). In both quiescent and activated NBs of control animals, pAKT expression was not detectable by immunohistochemical methods. However, upon *Plp-CreER^T2*-mediated recombination, peripheral nerve cells prominently expressed pAKT, indicating efficient activation of PI3K/AKT signalling pathway upon *Pten* deletion (Fig. 6b). Remarkably, NBs of *Pten* cKO animals displayed an overt expansion of nerve-derived cells in the wound (Fig. 6c). Quantification revealed an approximately two-fold increase in the number of p75⁺ and lineage-traced cells in the wound bed of *Pten* cKO mice (Fig. 6d, e).

Normal wound healing in healthy mice appears to be naturally optimised and it has been notoriously hard to experimentally improve this process[2,43]. Indeed, the increase in the number of nerve-derived cells in the wound of *Pten* cKO mice did neither lead to a significant acceleration of the wound closure D7 (86.3 ± 5.0% covered by epidermis in control vs. 87.2 ± 5.1% in *Pten* cKO) nor did it alter the area of HPE (0.250 ± 0.02 vs. 0.278 ± 0.03 mm²), wound width (2.12 ± 0.09 vs. 1.90 ± 0.10 mm) or length of wound epithelium (1.81 ± 0.06 vs. 1.68 ± 0.08 mm), measured at the same time point (Fig. 6f–i).

Interestingly, however, the area of α-SMA staining in the wound bed significantly increased upon *Pten* cKO, with myofibroblasts occupying around 47% of the total granulation tissue area in *Pten* cKO wounds, compared to 34% in control settings (Fig. 6j, k). Importantly, consistent with our finding that TGF-β signalling was reduced upon depletion of injury-activated glia (Fig. 5l, m), we found that increasing the number of activated glial cells led to enhanced TGF-β signalling in cells of the

granulation tissue. Indeed, at D5, *Pten* cKO wounds showed an increased number of cells with nuclear staining for pSMAD2 compared to control ($22.2 \pm 2.9\%$ vs. $14.2 \pm 1.2\%$, respectively) (Fig. 6l). In sum, conditional loss of *Pten* caused an expansion of injury-activated glia in the regenerating skin, which was associated with activation of TGF-β signalling and a concomitant

increase in α-SMA expression in the wound. These findings demonstrate a role of activated peripheral glia in regulating the number of myofibroblasts during wound healing.

**Glia promote myofibroblast formation via TGF-β signalling.** To directly demonstrate the capacity of peripheral glia to induce

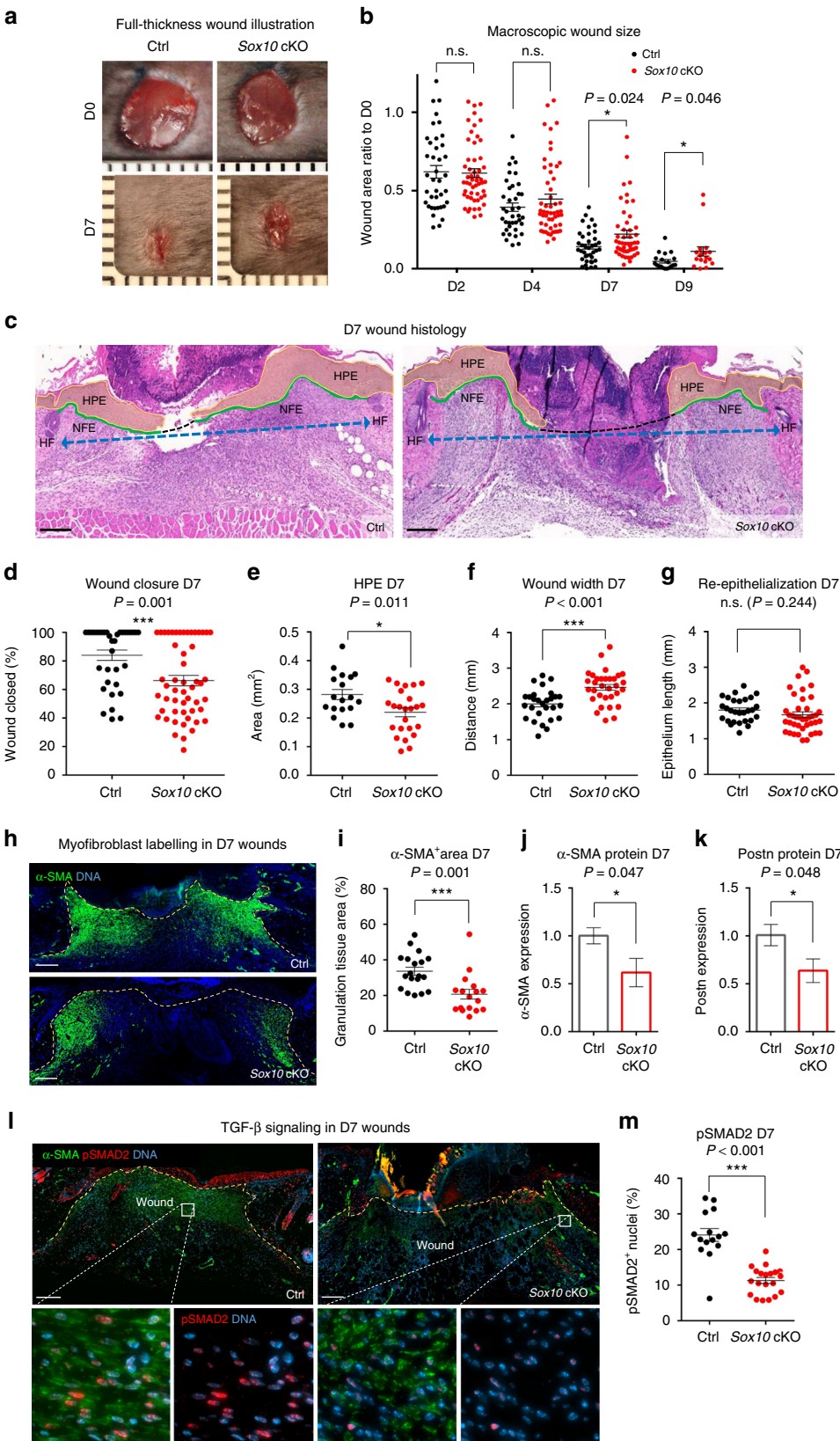

myofibroblast differentiation from dermal fibroblasts, we established a co-culture system of human fibroblasts (HuFs) and murine sciatic nerve explants (SN Exp) obtained from *Dhh-Cre; tdTomato* animals, as depicted schematically in Fig. 7a. In this system, cells of murine origin can be distinguished from unlabelled HuFs by virtue of their tdTomato expression. Intriguingly, upon dissection and culture alone, glia in peripheral nerves appear to undergo processes very much resembling those observed in vivo upon wounding. Indeed, murine tdTomato[+] Sox10[+] glial cells in the SN Exp—fibroblast co-culture system de-differentiated through expression of p75 and c-Jun and spontaneously emigrated from the nerve explants (Fig. 7b). To assess the effect of these activated glial cells on myofibroblast differentiation, and to decipher a potential implication of TGF-β signalling in this process, we performed α-SMA staining of HuFs cultured with or without murine SN Exp, in the presence or absence of the TGF-β signalling inhibitor SB431542 (SB43)[44] (Fig. 7c). Of note, co-culture with SN Exp significantly increased the number of α-SMA[+] HuFs when compared to cultures without peripheral nerves (16.1 ± 1.2% vs. 22.3 ± 1.8%, respectively) (Fig. 7d). Moreover, α-SMA expression was higher in close proximity to the murine tissue as compared to areas in the same culture dish but further away from the SN Exp, further demonstrating the α-SMA-inducing effect of nerve-derived cells (Fig. 7e, f). Importantly, nerve-dependent induction of α-SMA expression in HuFs was abolished upon TGF-β signalling blockade (Fig. 7d, f). Thus, activated glial cells induce myofibroblast formation through paracrine activation of TGF-β signalling.

## Discussion

In the present study, we demonstrate a novel role of injury-activated peripheral glia during wound healing of the skin. Lineage-specific genetic tracing, visualised among others by tissue clearing techniques combined with 3D-imaging, revealed that glial cells upon skin injury de-differentiate to express markers characteristic for repair Schwann cells formed in response to nerve injury[18,29] and start proliferating. Intriguingly, activated glia then disseminate from the injured nerves into the granulation tissue, where they support non-neural tissue repair of the wounded skin by paracrine signalling. In particular, depletion of injury-activated glia impaired TGF-β signal activation and the appearance of myofibroblasts in the skin; in contrast, genetically induced expansion of injury-activated glia in vivo as well as their co-culture with fibroblasts promoted TGF-β signalling and, consequently, myofibroblast formation.

It is known that injury to peripheral nerves is associated with peripheral glia activation and de-differentiation[29]. However, the signalling event triggering this process remains largely elusive.

Although certain extracellular factors including interleukin (IL)-6[45] and tumour necrosis factor (TNF)-α[46] have been implicated in Schwann cell activation, culture of dissected peripheral nerves without addition of these factors was sufficient to enhance p75 and c-Jun expression in glia. Likewise, co-culture with other cell types was not required to obtain glial activation in dissected peripheral nerves, speaking against a major involvement of skin cells, such as keratinocytes, dermal fibroblasts, or immune cells. Together with the correlation between loss of axonal NF and glial de-differentiation that we measured in activated NBs of wounded skin, our data suggest that impaired physical integrity of peripheral nerves might be sufficient to trigger glial activation.

Activated peripheral glia induced by nerve injury have been reported to acquire a plastic phenotype reminiscent of Schwann cell precursors found during embryonic development[29]. Schwann cell precursors appear to be multipotent and, apart from glia, can give rise to endoneurial fibroblasts[47], melanocytes[12], parasympathetic neurons[48,49], and mesenchymal cells[50]. Moreover, after environmental challenge, PNS glia in the adult can also give rise to melanocytes[12], neurons[51], and tooth mesenchymal cells[50]. Therefore, we had to consider the possibility that activated glia in the wound bed structurally contribute to scar formation through differentiation into non-glial lineages. However, lineage tracing combined with marker expression analyses did not provide significant evidence for injury-activated glia adopting alternative fates. In particular, traced cells neither generated melanocytes nor myofibroblasts, which are both cell types that can be produced by neural crest cells, the precursors of Schwann cells during development. Instead, in the undisturbed 3D context of the healing wound, injury-activated glia displayed lineage-restriction in vivo and continued to express peripheral glial lineage markers even after prolonged time points after injury. This is reminiscent of Schwann cell precursors in the Mexican salamander axolotl (*Ambystoma mexicanum*) that maintain Schwann cell identity after limb amputation while promoting regeneration[52] and of astroglia in the mammalian CNS that are also activated in response to injury and contribute to tissue repair, but exclusively give rise to astrocytes unless genetically manipulated[53].

Rather than a direct contribution of activated peripheral glia to cell types constituting the wound bed, our study demonstrates a cell non-autonomous effect on these cells during wound healing. Expression profiling of genetically traced peripheral glial cells either from intact or wounded skin demonstrated a significant enrichment for expression of various ligands and gene products destined to the extracellular space, representing a putative secretome of Schwann cells upon injury. This secretome includes gene products reported to affect multiple pathways, such as ECM remodelling, angiogenesis, chemotaxis of immune cells, or TGF-β signalling, which are all biological processes well known to

**Fig. 5** Depletion of injury-activated glia in the granulation tissue impairs wound healing. **a** Representative macroscopic illustration of wound healing in Ctrl and *Sox10* cKO animals at D0 and D7. **b** Longitudinal macroscopic quantification of individual wound areas at D2, D4, D7, and D9 in Ctrl and *Sox10* cKO animals in relation to their respective initial size at D0. Data show a significant delay in wound closure at D7 upon *Sox10* cKO. **c** H&E-stained sections of Ctrl and *Sox10* cKO D7 wounds used for morphometric analysis of percentage of wound closure (length of newly formed epithelium (NFE)/length of NFE + length of gap between edges of wound epithelium (black dotted line) × 100), area of HPE, wound contraction (distance between wound border HFs (blue dotted line)), and re-epithelialisation (length of NFE). Analysis of D7 wound sections for the percentage of **d** wound closure, **e** area of HPE, **f** wound width, **g** re-epithelialisation. **h** Immunolabelling of myofibroblasts with α-SMA (green) in Ctrl and *Sox10* cKO skin wound sections at D7. Dashed line delimits granulation tissue. **i** Quantification of the percentage of α-SMA[+] area of the granulation tissue at D7 shows a reduction of the area occupied by myofibroblasts upon *Sox10* cKO. Western blot quantification for **j** myofibroblast marker α-SMA and **k** Periostin (Postn) of D7 whole-wound protein extracts from Ctrl and *Sox10* cKO animals. **l** Immunolabelling of α-SMA (green) and pSMAD2 (red) in Ctrl and *Sox10* cKO skin wound sections at D7. Boxed regions in the dermis are shown in the insets at higher magnification as pSMAD2 single channel and merged with α-SMA channel. **m** Quantification of highly pSMAD2[+] cells per α-SMA[+] area of the granulation tissue at D7. Data are represented as mean ± SEM of *N* = 10, *n* = 39 (Ctrl), *N* = 13, *n* = 49 (*Sox10* cKO) (**b**); *N* = 9, *n* = 35 (Ctrl), *N* = 13, *n* = 52 (*Sox10* cKO) (**d**); *N* = 5, *n* = 18 (Ctrl), *N* = 6, *n* = 24 (*Sox10* cKO) (**e**); *N* = 7, *n* = 28 (Ctrl), *N* = 8, *n* = 32 (*Sox10* cKO) (**f**); *N* = 7, *n* = 28 (Ctrl), *N* = 10, *n* = 40 (*Sox10* cKO) (**g**); *N* = 5, *n* = 19 (Ctrl), *N* = 5, *n* = 17 (*Sox10* cKO) (**i**); *N* = 3, *n* = 6 per group (**j**, **k**); *N* = 8, *n* = 15 (Ctrl), *N* = 10, *n* = 20 (*Sox10* cKO) (**m**). Scale bars, 200 μm

influence wound healing[2]. To test whether any of these processes were significantly impacted by our cell population of interest, we genetically ablated injury-activated glia by preventing their expansion into the wound bed using lineage-specific conditional deletion of the transcription factor *Sox10*, a factor known to control neural crest cell survival and glial lineage specification[54].

Preventing dissemination of glial cells into the wound bed delayed wound healing and, notably, interfered with wound contraction and proper myofibroblast formation, which was associated with decreased TGF-β signalling in a non-cell autonomous manner. These data are in line with previous reports showing a direct link between TGF-β, granulation tissue

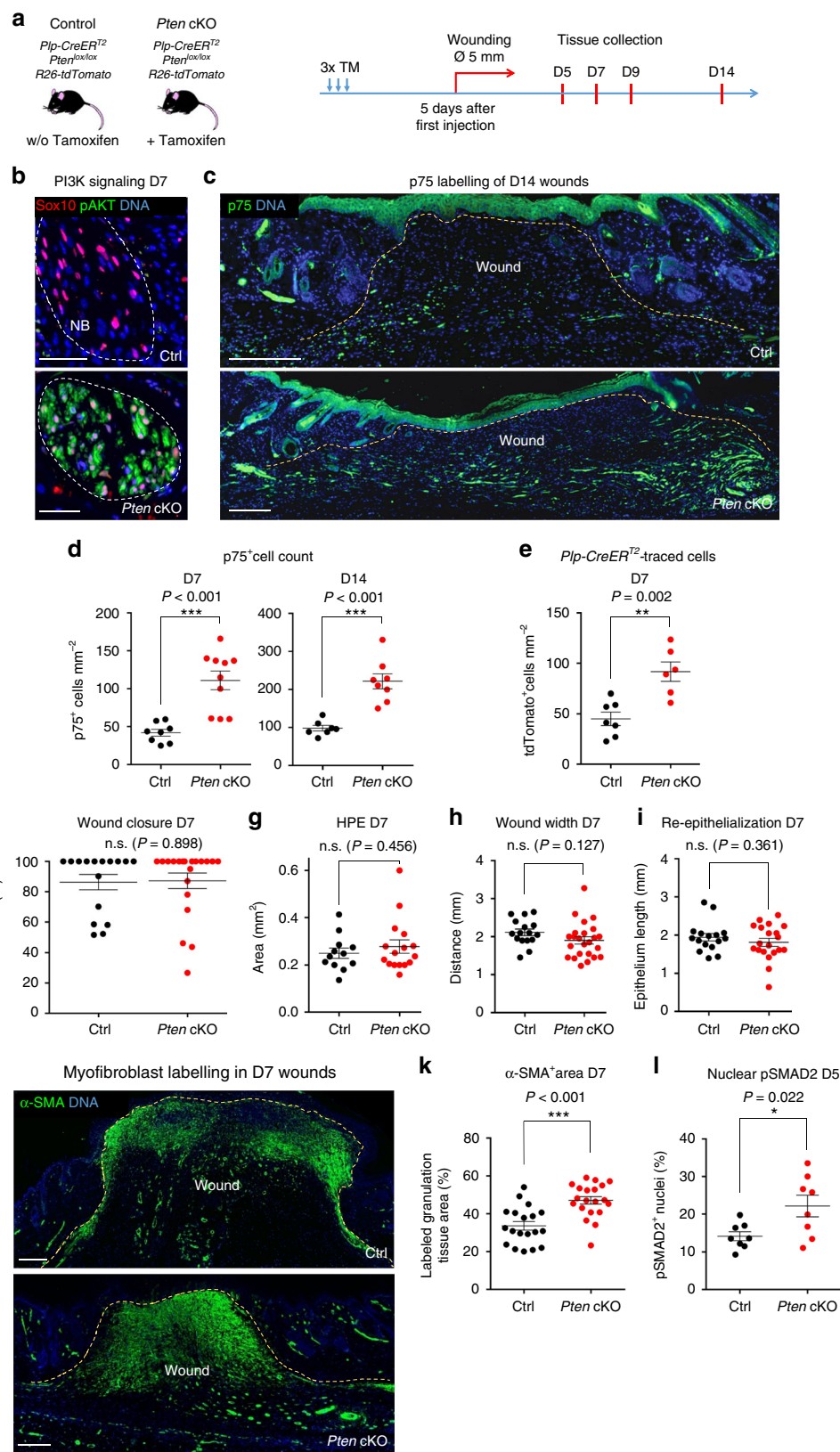

myofibroblasts, and wound contraction in vivo[55,56]. Accordingly, expansion of injury-activated glia in the wound bed, achieved by genetic ablation of *Pten*, led to increased paracrine TGF-β signalling and myofibroblast counts. Furthermore, genetically traced glia emigrating from dissected peripheral nerves promoted a fibroblast to myofibroblast transition in a newly established co-culture system. This effect was fully blocked by addition of a TGF-β signalling inhibitor, identifying TGF-β signalling as a mediator of the paracrine effect exerted by injury-activated glia.

However, besides the activin/inhibin ligand Inhba, our study did not provide evidence for specific TGF-β ligand expression by injury-activated glia. Rather, it pointed to modulation of ligand availability through ECM remodelling and TGF-β post translational modification, likely conferred by factors known to regulate these processes and present in the glia secretome, such as *Lox*, *Loxl2*, and *Plod2* and *Bmp1* (references in Supplementary Tables 2 and 5). This is consistent with previous reports indicating that non-myelinating Schwann cells have the potential to secrete molecules regulating latent TGF-β activation[57]. Although many other factors of the glia secretome characterised in our study have been functionally associated before with wound healing (see references in Supplementary Tables 1–6), their contribution to the paracrine effects by injury-activated glia remains to be elucidated. Notably, the amount of Periostin, a known inducer of myofibroblast formation and modulator of several other processes involved in wound healing[41], was decreased in the wound bed upon genetic depletion of injury-activated glia. Given the many distinct classes of factors present in the glia secretome, it is conceivable that injury-activated glia take part in processes of skin wound healing other than promoting myofibroblast formation. Indeed, loss of injury-activated glia also affected the area of newly formed, HPE. However, expansion of peripheral glia in the wound bed did not influence this process, presumably because it is naturally optimised in murine skin wound healing[43]. Nonetheless, our data speak for multiple effects of injury-activated glia and point to a therapeutic potential of these cells and/or of their secretome in human wound healing.

Our work is in line with other recent reports that showed crucial roles of peripheral nerve-derived cells in tissue regeneration and repair, both in amphibians and mice[9,58,59]. For instance, specialised repair glial cells forming after nerve injury were found to guide axons and restore innervation[21,28]. Ablation of Sox2-expressing cells, which are located in nerve terminals in the skin, peripheral nerves, and dermal papillae in HFs, impaired mouse skin wound healing and digit regeneration[9,59]. Likewise, transplantation of Schwann cell precursors isolated from sciatic nerves elicited a positive effect on digit tip regeneration by promoting mesenchymal cell proliferation and bone regeneration via secretion of Oncostatin M and PDGF-AA[59]. Together with our study showing their influence on TGF-β-mediated myofibroblast formation and skin wound closure, peripheral glia emerge as

important players in tissue regeneration. Possibly, this involves interaction of activated glia with other stem and progenitor cells types. In fact, Schwann cell precursors purified from neonatal sciatic nerve can stimulate self-renewal of skin-derived precursors[59], a mesoderm-derived mesenchymal cell population able to reconstitute the dermis[60,61]. It remains to be shown, though, whether the nerve-derived cell populations reported in the literature to have regenerative potential represent distinct cell types or one dynamic cell population eliciting distinct paracrine responses to injury depending on the type of injury or the organ affected. Likewise, it will be highly relevant to address whether injury-activated glia are also implicated in other tissues for which regeneration was shown to depend on innervation[62], including heart and skeletal muscle[63–65].

## Methods

**Mouse strains.** *Dhh-Cre*: Tg(Dhh-cre)1Mejr[15]–a transgene consisting of a Cre coding sequence inserted at the start codon of a *Dhh* genomic clone (strain of origin: FVB/N). *Plp-CreER^T2*: Tg(Plp1-cre/ERT2)1Ueli[11]–a transgene consisting of the mouse promoter of the *Plp* gene driving the expression of a TM inducible Cre recombinase (strain of origin: (C57BL/6 × DBA/2)F1). *Pten^lox*: Pten^tm1Hwu[66]–a floxed allele of *Pten* allowing the suppression of its phosphatase activity upon Cre-mediated recombination (strain of origin: 129S4/SvJae). *Sox10^lox*: Sox10^tm7.1(Sox10)Weg[54]–a floxed allele of *Sox10* allowing the conditional KO of the gene upon Cre-mediated recombination (strain of origin: 129P2/OlaHsd). *tdTomato*: Gt(ROSA)26Sor^tm14(CAG-tdTomato)Hze[67]–a tdTomato sequence preceded by a loxP-flanked STOP cassette targeted to the Rosa locus. This allows Cre reporting and subsequent lineage tracing by following tdTomato-expressing cells (strain of origin: (129S6/SvEvTac × C57BL/6NCrl)F1). *Tyr-CreER^T2*: Tg(Tyr-cre/ERT2) 13Bos[24]–a transgene consisting of 5.5 kb fragment of the Tyr promoter and a 3.6 kb fragment of the Tyr enhancer driving expression of a TM inducible form of Cre recombinase.

**Experimental mouse lines.** For genetic tracing of neural crest-derived population *Plp-CreER^T2*; *tdTomato*, *Dhh-Cre*; *tdTomato*, and *Tyr-CreER^T2*; *tdTomato* transgenic mouse lines were used. For the ablation of neural crest-derived cell population we used *Plp-CreER^T2*; *Sox10^lox/lox*; *tdTomato* transgenic mice. For the expansion of neural crest-derived cell population *Plp-CreER^T2*; *Pten^lox/lox*; *tdTomato* transgenic mice were used. All animal experiments have been approved by the veterinary authorities of Canton of Zurich, Switzerland, and were performed in accordance with Swiss law on the care, welfare, and treatment of animals.

**Genotyping.** DNA extracted by standard procedures from tissue samples was analysed by PCR, using gene-specific primers (Supplementary Table 7).

**Experimental animals.** In our wound healing experiments, we used animals bred in-house, aged between 2 and 6 months of several combined transgenic lines, with their skin hair cycle in the late anagen to telogen phases. Due to the various strains of origin of our transgenes, the genetic background is empirically considered mixed. Cell population characterisation by immunolabelling was performed on a cohort mixed gender. Littermates were evenly distributed across control and cKO experimental groups. All quantifications were performed on tissue samples obtained from males only. The experimenters were not blinded and were aware of the genotype of the animals prior to experimentation. No statistical method was applied to predetermine the same size of experimental groups. Sample sizes were large enough to measure the effect size. Appropriate animal health status, in accordance with our animal experimentation protocol, was used for exclusion

**Fig. 6** Increase of injury-activated glial cell numbers in the wound bed boosts myofibroblast differentiation through TGF-β activation. **a** Scheme of the experimental strategy for *Plp-CreER^T2*-mediated glial cell expansion in our murine wound healing model through *Pten* cKO. **b** Immunolabelling of pAKT (green) and Sox10 (red) in NB of skin wound sections at D7 shows a significant increase in PI3K signalling upon *Pten* cKO. **c** Immunolabelling of p75 (green) in Ctrl and *Pten* cKO skin wound sections at D14. **d, e** Quantification of p75$^+$ and tdTomato$^+$ cells per mm$^2$ of granulation tissue shows a significant increase in p75$^+$ cells at D7 and D14 and traced cells at D7 upon *Pten* cKO. Quantification of D7 wound sections of the percentage of **f** wound closure, **g** area of HPE, **h** wound contraction, **i** re-epithelialisation reveals no difference in these parameters. **j** Immunolabelling of α-SMA (green) in Ctrl and *Pten* cKO skin wound sections at D7. **k, l** Quantification of the percentage of α-SMA$^+$ area of the granulation tissue at D7 and of the percentage of pSMAD2$^+$ cells per α-SMA$^+$ area of the granulation tissue of skin wounds at D5 shows increase of both myofibroblast-occupied area and canonical TGF-β signalling in *Pten* cKO skin wounds. Note that control data sets in Fig. 6k and Fig. 5i are the same because the experiments were performed at the same time. Data are represented as mean ± SEM of $N = 2$, $n = 8$ (Ctrl D7), $N = 2$, $n = 8$ (Ctrl D14); $N = 3$, $n = 10$ (*Pten* cKO D7), $N = 2$, $n = 8$ (*Pten* cKO D14) (**d**); $N = 4$, $n = 7$ (Ctrl), $N = 2$, $n = 6$ (*Pten* cKO) (**e**); $N = 4$, $n = 16$ (Ctrl), $N = 5$, $n = 20$ (*Pten* cKO) (**f**); $N = 3$, $n = 12$ (Ctrl), $N = 4$, $n = 16$ (*Pten* cKO) (**g**); $N = 4$, $n = 16$ (Ctrl), $N = 6$, $n = 24$ (*Pten* cKO) (**h**); $N = 4$, $n = 16$ (Ctrl), $N = 5$, $n = 20$ (*Pten* cKO) (**i**); $N = 5$, $n = 19$ (Ctrl), $N = 5$, $n = 20$ (*Pten* cKO) (**k**); $N = 3$, $n = 8$ (Ctrl), $N = 2$, $n = 8$ (*Pten* cKO) (**l**). Scale bars, 50 μm (**b**), 200 μm (**c**, **j**)

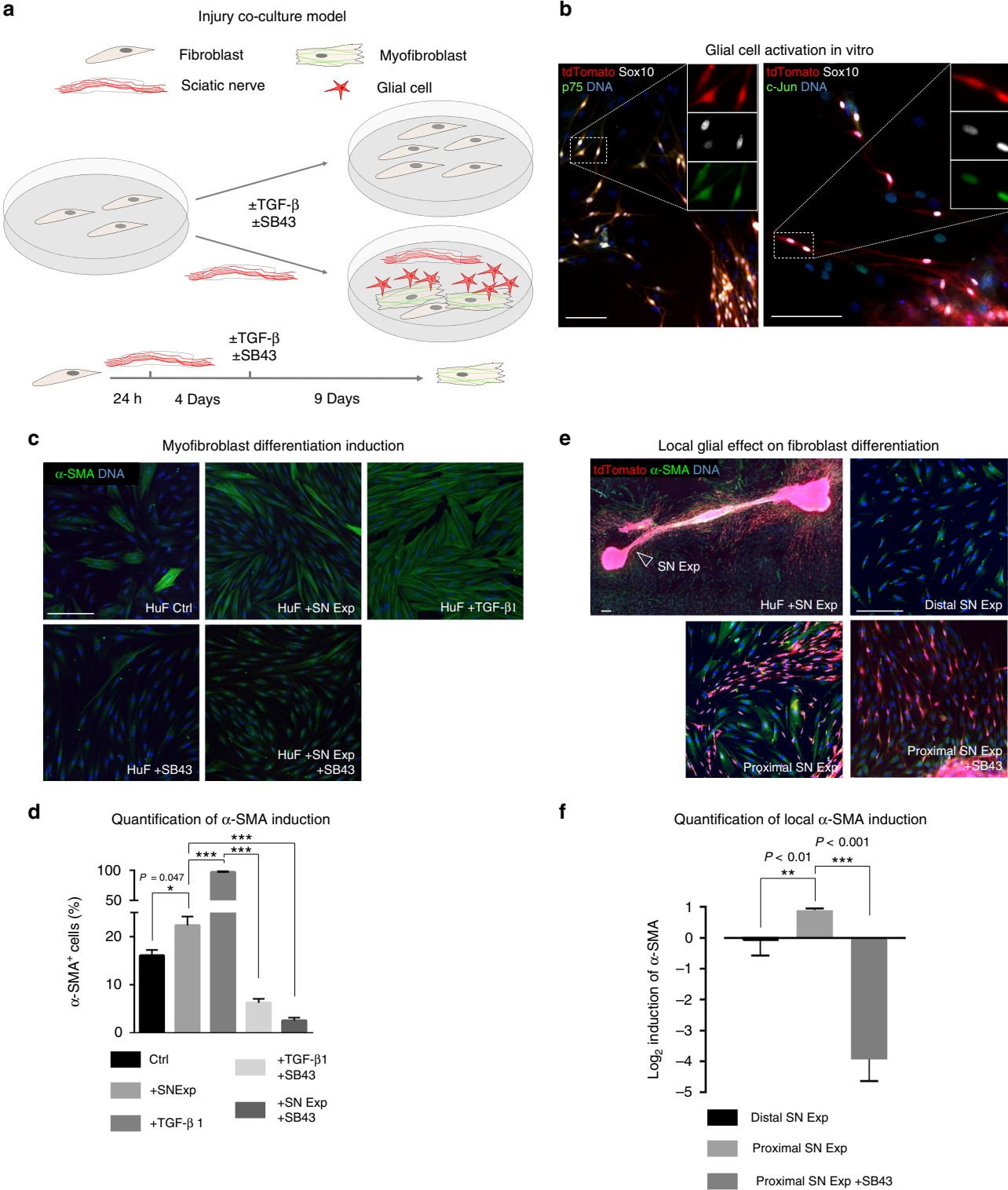

**Fig. 7** Activated glia from SN Exps promote myofibroblast differentiation in a TGF-β signalling-dependent manner. **a** Experimental scheme of HuF cultured in the presence of murine *Dhh-Cre; tdTomato* SN Exp with optional pharmacological modulation of TGF-β signalling pathway. **b** Immunofluorescence of SN Exp show tdTomato-traced cells (red) expressing markers of activated glia, such as Sox10 (white), p75 (green), and c-Jun (green). Note that similar results were obtained even in the absence of fibroblasts. **c** Immunolabelling of α-SMA[+] (green) HuF upon SN Exp co-culture, TGF-β1 treatment or SB43-mediated TGF-β signalling blockade and **d** respective quantification of α-SMA induction for each culture condition. ***$P < 0.001$ compared to Ctrl. **e** Immunolabelling of local α-SMA[+] induction in tdTomato[−] HuF in the direct vicinity of the SN Exp compared to areas distant from the SN Exp (one field of view from nearest tdTomato[+] cell using 10× objective) of the same well in combination with SB43-mediated TGF-β signalling blockade. **f** Respective quantification of local α-SMA induction for each culture condition. Data are represented as mean ± SEM. Scale bars, 100 μm (**b**), 300 μm (**c, e**).

criteria. No randomisation of the animals was used as experimental group assignment was primarily dependent on the appropriate animal genotype.

**Cre-mediated allele recombination.** cKO of *Sox10* using the *Plp-CreER^T2* driver was performed using i.p. injection of a 10 mg ml$^{-1}$ TM solution in sunflower oil/EtOH 9:1. Animals were injected daily with 200 µl TM for 5 consecutive days for a total of 10 mg TM per animal. To obtain efficient depletion of the Sox10-expressing population in the skin, cKO of *Sox10* was performed 3 weeks prior to injury.

*Pten* cKO was performed using a 3-day treatment of daily i.p. injection of 10 mg ml$^{-1}$ TM solution. Operative procedure was performed 5 days after the initial injection.

**Excisional wounding of murine skin.** Following a protocol approved by the Zurich cantonal veterinary office, general anaesthesia of the animals was induced by 5% Isoflurane in 100% $O_2$ and subsequently maintained using 3% Isoflurane. The back skin was shaved, thoroughly cleaned, and disinfected prior the surgery using antibacterial soap solution and 75% EtOH solution. Preoperative analgesia was applied using a subcutaneous injection of Buprenorphine (0.1 mg kg$^{-1}$). Four circular full-thickness excisional wounds of 5 mm of diameter were generated on the lower back skin of each animal, two on each side, 1 cm of the midline of the animal and roughly 2 cm apart from each other. Post-operative analgesia was performed by a 5-day treatment of Buprenorphine through the drinking water. Wounds were allowed to heal without dressing.

**Macroscopic wound size measurement.** Wound area was determined for each wound by quantification of wound area based on pictures taken from day 0 to day 14 post surgery. The area at different time points after wounding were then reported to the original area value at day 0 for each wound.

**Histological analysis of murine skin samples.** At the day of sample collection, animals were euthanised using $CO_2$ and skin samples were excised post-mortem, fixed in a buffered 4% formaldehyde solution at 4 °C, overnight for samples to be embedded in paraffin. For samples to be embedded in tissue freezing medium or subjected to clearing 4 h fixation at 4 °C was performed. Paraffin blocks were processed into sections of 5 µm while cryoblocks were processed into 12 µm sections. For immunofluorescence, protocols described elsewhere were applied. Briefly, sections were deparaffinised and subjected to an antigen retrieval step using citrate buffer (S2369, Dako). Primary antibodies (Supplementary Table 8) were applied in blocking buffer (2% BSA in PBS and 0.05% Triton X-100) overnight at 4 °C and visualised using secondary antibodies (Supplementary Table 9) in blocking buffer for 1 h at room temperature. For pERK, c-Jun and pSMAD2 staining TSA amplification kit was used (ThermoFischer). Hoechst 33342 (14533, Sigma-Aldrich) usually was used as nuclear counterstain at a 1 µg ml$^{-1}$ working concentration, unless stated otherwise. Haematoxylin and Eosin (H&E) and Herovici-stained slides were processed according to standard protocols. Immunofluorescence-stained, H&E-stained, and Herovici-stained sections were imaged using either a DMI 6000B microscope (Leica) or an Axio Scan.Z1 slide scanner (Zeiss). Image analysis and quantifications were performed using ImageJ 1.49 (National Institutes of Health, USA) and ZEN (Zeiss) imaging software. For pSMAD2 quantification three high magnification fields (HMF) per wound were randomly selected within the area positive for α-SMA, percentage of nuclei positive for pSMAD2 was quantified and averaged.

**Morphometric analysis.** Morphometric analysis was performed on 5 µm sections obtained from the middle of D7 wounds and stained with H&E. Measurements of wound closure, area of HPE (area of epidermis measured from the point outside of the wound, where it starts to become thicker than normal epithelium), distance between wound border HFs (contraction assessment), and length of newly formed wound epithelium (re-epithelialisation) were carried out using ImageJ as explained in Fig. 5c and as previously described[68].

**Expression profiling of genetically traced cells.** The skin of three non-wounded control *Dhh-Cre; tdTomato* animals and the 7-day-old wounds of their three littermates were collected post mortem and pooled in two groups. Subsequently, the tissue samples were mechanically chopped using a disposable scalpel and further dissociated to a single cell suspension using 0.25 mg ml$^{-1}$ Liberase DH Research Grade (05401054001, Roche) in RPMI 1640 (42401, Life Technologies) for 60 min at 37 °C with gentle rocking followed by a treatment with 0.55 mg ml$^{-1}$ Dispase II (17105, Life Technologies) and 0.2 mg ml$^{-1}$ DNase I (10104159001, Roche) for 20 min at 37 °C. The cell suspension was then passed through a 40 µm cell strainer and sorted for tdTomato positivity using FACSAria III (BD Biosciences) laser. Sorted cells were directly placed into RNA lysis buffer for subsequent RNA isolation using RNeasy Micro Kit (74004, Qiagen). Total RNA obtained from three control and three wounded skin samples (pools from four wounds) was then amplified using Ovation Single Cell RNA-Seq Multiplex System (0342-32-NUG, NuGEN) and subjected to sequencing on Illumina Hiseq 2500 platform at the Functional Genomics Center Zurich (http://www.fgcz.ch/). Differential gene expression analysis was performed using a minimum fold change of 1.5 and a False

Discovery Rate inferior to 0.05. Gene ontology network analysis was performed with MetaCore (Thomson Reuters).

**SN Exp co-culture and immunofluorescence.** Fibroblasts used in this study were derived from a human foreskin sample[69] after obtaining permission from the Ethics Commission of the Canton Zurich and after informed consent given by parents. For nerve explant co-cultures human dermal fibroblasts were seeded in 12-well plates at $7 \times 10^3$ cells cm$^{-2}$ density and cultured for 24 h in 10% FBS-supplemented DMEM. After 24 h the media was changed to 1% FBS DMEM and sciatic nerve fragments were placed in the middle of the well at low media volume (300 µl/well). Sciatic nerves were isolated from adult *Dhh-Cre; tdTomato* mice and processed as previously published[70]. Briefly, sciatic nerves were dissected under sterile conditions–nerves were cut in 2–3 mm pieces followed by careful epineurium and perineurium removal using fine forceps and binoculars. These fragments can be used as explants and passaged. After 4 days of co-culture fresh DMEM/F12 media containing 1% FBS only or combinations of supplements was added for final concentration of 10 ng ml$^{-1}$ TGF-β1 (100-21, Peprotech) and 10 µM SB 431542 (301836-41-9, Tocaris). DMSO was used as vehicle control. Fresh supplements were added to the media every day for 9 days.

For immunofluorescence cells were fixed for 6 min with 4% formaldehyde in PBS and blocked for 45 min (3% BSA, 0.1% Triton X-100 in PBS). Cells were incubated with primary antibodies (Supplementary Table 8) overnight at 4 °C (1% BSA in PBS) and for 1 h at room temperature with secondary antibodies (Supplementary Table 9) and Hoechst 33342 (PBS). For quantification, in each well cells positive for α-SMA with visible stress fibres were counted in at least four 1 mm$^2$ HMFs using a DMI 6000B microscope (Leica) and Image J software (National Institutes of Health, USA). Cells were quantified either distally from the explant (at least 1 mm from the migrating explant cells) or proximally–with migrating cells present in HMFs.

**Protein isolation and western blotting.** For wound protein isolation two D7 wounds were pooled for each sample. Wound tissue was transferred into RIPA buffer (89900, Thermo Scientific) containing Halt™ Phosphatase and Protease Inhibitor Cocktail (78440, Thermo Scientific) and tissue was homogenised using tissue homogeniser (Polytron) and subsequently sonicated three times for 10 s using ultrasonic homogeniser (Bandelin). SDS–PAGE was carried out on 4–20% Mini-PROTEAN TGX Gels (456-1094, Bio Rad). Primary antibodies (Supplementary Table 8) were applied in Odyssey blocking buffer (927–40000, LI-COR Biosciences) overnight at 4 °C and secondary antibodies (Supplementary Methods Table 9) for 1 h at room temperature. Blots were scanned and quantified using Odyssey imaging system (LI-COR Biosciences). Quantified band intensities were normalised using β-Actin and plotted on the graph. Full size scans of the blots are presented in Supplementary Fig. 5.

**Statistical analyses.** Statistical analysis was performed and graphs created using Prism 6 software (GraphPad Software Inc., San Diego, CA). Variation estimates were not performed or compared for each data sets. Statistical significance between two groups was calculated using unpaired Student's t-test for normally distributed data sets. For wound area measurements in Fig. 5b nonparametric Mann–Whitney test was used.

**Data availability.** The data sets generated during and analysed during the current study are available from the corresponding author on reasonable request. RNA sequencing data have been deposited to the European Nucleotide Archive under the accession code PRJEB22372.

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

## Acknowledgements

We would like to acknowledge the microscopy core facility (ZMB), the Functional Genomics Center Zurich (FGCZ), as well as the histology and animal facility personnel for their help with data acquisition, sample processing, and animal management, respectively. We would also like to thank Dr. Phil Cheng from the University Hospital Zurich for the help with graphical data representation of Fig. 3, Dr. Maria Antsiferova from ETH-Zurich for the help with wound morphometric analysis, and Khanh Huynh from the University of Zurich for assistance with some experiments. The *Dhh-Cre* mouse line was kindly provided by Prof. Dies Meijer, University of Edinburgh, UK. This project was supported by the University Medicine Zurich (SKINTEGRITY), Swiss National Science Foundation, notably the National Research Programme (NRP63) "Stem Cells and Regenerative Medicine" and the Scientific Exchange Programme (SCIEX-NMS[CH]).

## Author contributions

J.D., V.P., U.R., and L.S. designed the experiments; V.P., J.D., and S.M.S. performed the experiments; V.P., J.D., and L.S. analysed the data. O.S. and S.W. provided expertise for the murine injury protocol; O.S., M.G., and S.W. initiated the project. J.D., V.P., and L.S. wrote the manuscript. M.W. and U.S. provided mouse lines. All authors made important comments to the manuscript.

## Additional information

**Competing interests:** The authors declare no competing financial interests.

