## [Peer Review File · Nature Communications]

Reviewers' comments:

Reviewer #1 (Remarks to the Author):

This is an interesting article showing a role for glia cells in skin wound healing. While a role for glial cells in scar formation in the nervous system has been established, this is the first demonstration that I am aware of for a role in skin repair.

The lineage and cell depletion studies show a role for glial cells in the repair process in a cell non-autonomous manner, and this data is the strongest part of the manuscript. The relative contribution of glial cells with other factors could be expanded upon in the discussion, as since healing can still proceed, albeit in a reduced manner, in the absence of these cells, there must be other sources of the relevant factors produced by glial cells.

Expression profiling identified a large number of differentially expressed genes, and while the authors suggest mediators, they do not provide direct evidence that TGF-beta is a mediator in the process. Studies that would directly deplete these factors for glial cells would more directly provide data to support this hypothesis.

Given the variability in healing parameters and overlap in finding between knockout and control wounds, the authors may wish to include additional analyses of wound healing as an additional verification for observed differences. Wound volume, strength, or matrix content could be considered.

Reviewer #2 (Remarks to the Author):

The manuscript reports that peripheral glial cells (dedifferentiated Schwann cells) play a crucial role in wound healing. By using genetic fate mapping experiments (involving two distinct Schwann cell-specific cre lines) and immunohistochemical analyses, the authors show that dedifferentiated Schwann cells disseminate from the nerve bundles into the wound area. Using FACS and RNA-Seq analyses, they identify TGFb as an important factor that these Schwann cells secrete, which helps in the wounding process by promoting myofibroblast differentiation. Notably, by depleting these dedifferentiated Schwann cells genetically, they show that the wounding process is impaired.

In general, this is a very well designed and comprehensive study with an impressive series of experiments and robust analytical approaches. The manuscript is clearly written and well organized. Conceptually, however, this work is similar to another 2 previously published studies as the authors themselves acknowledge in their Introduction and Discussion. In the first study, Johnston and colleagues (Stem cell reports 2013, Ref 9) also showed using similar genetic fate mapping strategy, that peripheral glia migrated to wound areas, and their genetic depletion led to impaired wound closure. In a subsequent recent study, the same group showed similar results, using another strategy of genetic depletion of the glial cells, involving mice carrying active diphtheria toxin A (Johnston et al., Cell Stem Cell 2016,

Ref 54). This present study by Parfejevs and colleagues, however, is complementary to the 2 other previous studies, and resolves one key issue: the identity of the glial cells that migrate to the wound area: the authors convincingly demonstrate that these cells are dedifferentiated Schwann cells, since they use 2 cre-line for their fate mapping and genetic depletion studies that are more closely targeted to cells of the Schwann cell lineage (Dhh-cre and Plp-Cre ERT2) than the Sox2-Cre lines used in the other 2 previously studies, which can also label Merkel cells or cells associated with nerve terminals. Through their elegant FACS and RNA-Seq studies, they also show that these cells could be important for myofibroblast differentiation by secreting TGFb.

The authors mention that peripheral neuropathies are known to severely impair the wound healing capabilities of the skin. This would suggest that the present mechanisms they are describing could be an important factor in this defective healing process. It would considerably strengthen this manuscript if the authors were to examine this in an adequate neuropathy model.

Minor points:

1. What does actually induce the de-differentiation of the Schwann cells: direct physical trauma to the nerve bundles, or some factors that are secreted in the wounding process? Can the authors discuss about this?
2. Why is the percentage of pSMAD2+ nuclei so different in the control mice from the Sox10 cKO mice (24.1%) and control mice from Pten cKO model (14.2 %)? This is a difference of close to 50%. If the techniques were standardised, they should be similar values. Is it because of strain differences? Can the authors comment?

Reviewer #3 (Remarks to the Author):

In this extremely interesting paper the authors explore the role of nerve-derived Schwann cells in the repair of skin following a wound. They show convincingly that Schwann cells, from apparently damaged nerves, enter the wound bed in large numbers where they contribute to the healing of the wound. They demonstrate that the recruited Schwann cells secrete factors likely important for the healing process and show that the inhibition of their migration to the wound site leads to minor defects in the repair of the wound. In particular, the Schwann cells appear to play a role in the activation of fibroblasts, which are important for the closure of the wound. This work thus makes a significant additional contribution to findings that Schwann cells have important roles in wound healing and tissue regeneration. I have the following minor comments that would need to be addressed to make the manuscript suitable for publication in Nature Communications.

1. In Figure 1b, it would help to identify the melanocytes that are red with an additional marker.
2. In Figure 1e, it is really hard to determine what the figures are showing. Some additional labelling would help.
3. In Figure 2a, the nerve further from the wound appears to be activated/injured? Is this

the case? And if so can this be commented on as to why this should be.

4. In Figure 2b, the GFAP staining appears axonal.

5. In Figure 2c, it is not clear whether the p75+ cells are cjun positive and Ki67+. These double-labelled cells should be quantified.

6. The first section of the discussion is poorly written. It is not made clear what is new from these studies or what has already been shown by others.

7. Skips have been proposed to have an important role in the production of Schwann cells following injury and in cancer. The role of these cells should be discussed in the context of this study.

Reviewer #4 (Remarks to the Author):

The manuscript by Parfejevs et al is very carefully executed, analyzed, presented and written. The main conclusion is timely and exciting. I find it a bit surprising that Schwann cells don't contribute to melanocytes during healing, but the results seem robust. Perhaps it is the model they use, and contribution is greater in cases of hyperpigmentation associated with healed wounds? I just have a few minor comments.

1. It is unclear what makes some nerve bundles active and other inactive. Is there a correlation between lesioned axon bundles and active bundles? It would be good if the authors could address this issue experimentally, or at least speculate.

2. In general, the genetic animal models are very superficially explained in the main result. I think that most readers would want to understand what models were used and how they were used. The authors need to include the full genotype of each mouse model the first time it is used and explain how it will be abbreviated in the rest of the manuscript. For example, it is unclear how Sox10 is conditionally deleted using floxed alleles and the same time used for lineage tracing. Information on tamoxifen concentrations, administration regime and time at analysis is not always explained.

3. Similar issues relates to markers used, such as CD31, CD45 and more. These are never even mentioned in the main text. All information on markers used and what cell types they are believed to stain needs to be included in the result section. Similarly, please explain how (what) hyperproliferation was quantified.

4. It would have been nice to see an ablation experiment instead of Sox10 cKO strategy for establishing the role of these cells for wound healing. Is it possible that depletion of Sox10 changes the fate of the cells so they are still there, just with a different phenotype? There are protocols for topical administration of hydroxytamoxifen, so a local ablation strategy should be feasible. It would be nice if the authors could address this either experimentally or by clarifying what is actually believed to be happening with the Schwann cells when Sox10 is deleted.

5. Ref 54, Johnston et al., although examined tip regeneration has used a similar strategy. It would be good if the present study could be put into this context to a greater extent in the discussion.

Dear Reviewers of our manuscript NCOMMS-16-26618-T

We are very pleased to have been given the opportunity to revise our study on the role of injury-activated glia in skin wound healing and to resubmit our work to *Nature Communications*. We are grateful for the very supportive comments by all of you and thank you for having taken the time and effort necessary to provide critical input on how our findings could be strengthened. We carefully considered your comments and recommendations and, despite your already very positive feedback, sought to address the issues raised by you by performing quite extensive additional experimental work. I hope that you will share our opinion that this considerably improved our manuscript. Specifically, may I draw your attention to the following point-by-point replies to your comments:

1) Reviewer #1, Point 1:

This is an interesting article showing a role for glia cells in skin wound healing. While a role for glial cells in scar formation in the nervous system has been established, this is the first demonstration that I am aware of for a role in skin repair.

The lineage and cell depletion studies show a role for glial cells in the repair process in a cell non-autonomous manner, and this data is the strongest part of the manuscript. The relative contribution of glial cells with other factors could be expanded upon in the discussion, as since healing can still proceed, albeit in a reduced manner, in the absence of these cells, there must be other sources of the relevant factors produced by glial cells.

We thank the reviewer for this positive assessment of our work. We fully agree that cells other than glia must be involved in the wound healing process, as we also state in the

Introduction. We now further extend our Discussion to make that point (p.19).

2) Reviewer #1, Point 2:

Expression profiling identified a large number of differentially expressed genes, and while the authors suggest mediators, they do not provide direct evidence that TGF-beta is a mediator in the process. Studies that would directly deplete these factors for glial cells would more directly provide data to support this hypothesis.

This is indeed an important issue and we highly appreciate this reviewer's suggestion to provide further evidence for TGF- β signalling being implicated in the main role of injury-activated glia during skin wound healing, namely to promote myofibroblast differentiation from fibroblasts.

Based on the secretome analysis presented in Figure 3, we don't think that TGF- β itself is directly provided by injury-activated glia. In our GO analysis, we have now expanded the relevant categories of differential gene product localizations (revised Figure 3d-f), but the main message remains as in our previous version of the manuscript and as discussed on p.18: injury-activated glia do apparently not display increased expression of specific TGF- β ligands. Rather, our data point to modulation of ligand availability through ECM remodelling and TGF- β posttranslational modification, likely conferred by multiple factors known to regulate these processes and present in the glia secretome, such as *Lox*, *Loxl2*, and *Plod2* and *Bmp1* (see references in Supplementary Table S2). In addition, as we now show in revised Figure 5, Periostin, a factor also known to stimulate myofibroblast differentiation, is not only expressed by injury-activated glia, but its concentration is reduced in the wound upon depletion of activated glia. Thus, although valid as such, the suggested direct depletion of a TGF- β factor in glia would presumably not affect the observed glia-mediated effect on fibroblasts and we likely would have to conditionally ablate multiple TGF- β modulating factors.

Therefore, in order to address the reviewer's very relevant point, we had to find an alternative way to quantitatively assess TGF- β -mediated effects of nerve-derived glia on fibroblasts. To this end, we developed a novel co-culture system involving human fibroblasts and peripheral nerves dissected from mice, in which glia were genetically labelled by tdTomato. Intriguingly, dissection and isolation of peripheral nerves was sufficient to trigger glia de-differentiation and their dissemination from nerves, closely mimicking the processes we observed upon skin wounding in vivo (see new Figure 7a, b). Importantly, in such co-cultures in the presence or absence of the TGF- β signalling inhibitor SB431542, we were able to demonstrate that nerve-derived injury-activated glia are able to stimulate

myofibroblast differentiation in a TGF- β signalling dependent manner. Two different set-ups were used to quantify our data: either we compared the number of α -SMA-expressing fibroblasts in control dishes vs dishes containing dissected peripheral nerves (new Figure 7c); or we counted the number of myofibroblasts in close proximity to peripheral nerve explants vs those found distal to nerve cells (new Figure 7d). In both cases, we could demonstrate that the presence of nerve-derived activated glia promotes myofibroblast differentiation and that this effect can be efficiently blocked by addition of a TGF- β signalling inhibitor.

3) Reviewer #1, Point 3:

Given the variability in healing parameters and overlap in finding between knockout and control wounds, the authors may wish to include additional analyses of wound healing as an additional verification for observed differences. Wound volume, strength, or matrix content could be considered.

Based on our experience, 3D reconstruction of wound samples to measure the wound volume would need an excessively large number of wounds given the variability found in the wound healing process (see e.g. the number of samples analysed in Figures 5 and 6), which is why we performed our experiments using multiple tissue sections from control wounds and wounds from genetically modified mice. Together with an expert in biomechanics, we are currently trying to develop methods for measuring differences in skin strength and other mechanical properties between normal wounds and wounds from genetically modified mice. Although preliminary experiments with control wound samples look promising, the methods are not yet established to a point where measuring wound strength of genetically distinct samples would be feasible in a reliable manner.

However, to verify our observations using additional methods, as suggested, we isolated protein from control and *Sox10* cko wounds and performed western blot analysis of α -SMA and Periostin, a known inducer of myofibroblast formation and modulator of several other processes involved in wound healing. The results show significantly reduced α -SMA and Periostin levels upon depletion of injury-activated glia (new Figure 5j and k), in line with the other data presented in this study.

4) Reviewer #2, Point 1:

In general, this is a very well designed and comprehensive study with an impressive series of experiments and robust analytical approaches. (...) This present study by Parfejevs and colleagues, however, is complementary to (...) other previous studies, and resolves one key issue: the identity of the glial cells that migrate to the wound area.

The authors mention that peripheral neuropathies are known to severely impair the wound healing capabilities of the skin. This would suggest that the present mechanisms they are describing could be an important factor in this defective healing process. It would considerably strengthen this manuscript if the authors were to examine this in an adequate neuropathy model.

We are very happy about this reviewer's overall very positive evaluation of our study. Regarding the use of an "adequate neuropathy model", we would like to clarify that conditional loss of Sox10 in peripheral glia has been shown to result in neuropathy, as we also clarify on p.9, citing the appropriate literature (Bremer et al., 2011). For your perusal, please also find movies of control and Sox10 cko littermates illustrating the point. Please also note that in our study determining the specific role of glia in wound healing, we had to find a model in which glia are present in the control setting and genetically depletable and expandable, respectively, in experimental settings. The reviewer's point is well taken, however, and for a follow-up story we are currently assessing the presence or absence of injury-activated glia in human biopsies of neuropathy patients.

5) Reviewer #2, Minor Point 1:

What does actually induce the de-differentiation of the Schwann cells: direct physical trauma to the nerve bundles, or some factors that are secreted in the wounding process? Can the authors discuss about this?

An important issue, also brought up by Reviewer #4, is how Schwann cell de-differentiation is triggered in wounds. Reports on peripheral nerve injury have implicated factors such as TNF- α and IL6 as potential inducers of glia activation, but in our newly established in vitro system (see our Point 2) above), addition of these factors had no significant effect (data not shown). Of note, in this system, dissection and isolation of peripheral nerves was sufficient to trigger glia de-differentiation and their dissemination from nerves, closely mimicking the processes we observed upon skin wounding in vivo. As shown in new Figure 7b, when placing acutely dissected peripheral nerves in culture, genetically traced glial cells started to express p75 and c-Jun and emigrated from the nerves, thus mimicking the hallmarks of injury-induced glia activation in skin wounds *in vivo*. This was also the case when placing isolated peripheral nerves directly on culture dishes rather than on fibroblasts (data not shown). As now discussed on p.16, we conclude that presence of other cell types (such as immune cells and other dermal cells) is apparently not an absolute prerequisite for glial cell activation; rather, direct physical trauma seems to be sufficient to trigger this process, although the exact molecular mechanism underlying glial cell activation remains to be

determined.

6) Reviewer #2, Minor Point 2:

Why is the percentage of pSMAD2+ nuclei so different in the control mice from the Sox10 cKO mice (24.1%) and control mice from Pten cKO model (14.2 %)? This is a difference of close to 50%. If the techniques were standardised, they should be similar values. Is it because of strain differences? Can the authors comment?

The difference can be explained by the different time points used for these experiments, as we indicate in the corresponding figures (Figure 5m, day 7; Figure 6l, day 5).

7) Reviewer #3, general remarks:

We thank the reviewer for his very positive feedback to our study.

8) Reviewer #3, Point 1:

In Figure 1b, it would help to identify the melanocytes that are red with an additional marker.

We now show in a revised Supplementary Figure 2 Dct expression in tdTomato-positive cells in non-wounded and (now also as overview) wounded skin of *Plp-CreERT2 :: tdTomato* mice.

9) Reviewer #3, Point 2:

In Figure 1e, it is really hard to determine what the figures are showing. Some additional labelling would help.

As suggested, we added further labelling in Figure 1e (and in the figures in general) to clarify genotypes and markers used, to indicate wounded vs non-wounded samples, structures shown, time points of analysis, etc.

10) Reviewer #3, Point 3:

In Figure 2a, the nerve further from the wound appears to be activated/injured? Is this the case? And if so can this be commented on as to why this should be.

As mentioned in our Point 5) above, we believe that direct injury to nerves induces glia cell activation. Peripheral nerve injury leads to axon retraction, which is why loss of neurofilament (NF) staining is associated with glia cell activation. The structural injury to a given nerve bundle could have occurred in a different spatial plane than captured by the tissue section used for the analysis. We now exchanged the panels of revised Figure 2 and

show sections on which we were able to perform all stainings at ones and to classify nerve bundles according to their p75/NF expression. As shown in new Figure 2a,b, there is a correlation between loss of NF and presence of p75, further supporting the hypothesis that glia activation is linked to direct physical trauma to nerves.

11) Reviewer #3, Point 4:

In Figure 2b, the GFAP staining appears axonal.

We thank the reviewer for having pointed this out to us. We repeated the staining and indeed found co-labeling with neurofilament-positive axons. Since other anti-GFAP antibodies displayed similar cross-reactivity, we removed the panels with GFAP, in which axonal staining could lead to misinterpretation of the data (see revised Figure 2). Given the reliable staining of activated glia with anti-p75, c-Jun and Ki67 antibodies and the concomitant loss of MBP expression, omitting GFAP in this figure does not hamper the conclusion that wounding results in emergence of de-differentiated glia.

12) Reviewer #3, Point 5:

In Figure 2c, it is not clear whether the p75+ cells are cjun positive and Ki67+. These double-labelled cells should be quantified.

We appreciate this reviewer's very valid suggestion to quantify the extent of de-differentiation observed upon wounding. The data now shown in new Figure 2e reveal that a large fraction of p75-positive cells within activated nerve bundles also express c-Jun and that many of these double-positive cells are proliferative. Likewise, we now also demonstrate in a new Supplementary Figure 1 that the vast majority of p75-positive glia that have emigrated from nerve bundles upon wounding express c-Jun and that many of these disseminated double-positive cells also express the proliferation marker Ki67.

13) Reviewer #3, Point 6:

The first section of the discussion is poorly written. It is not made clear what is new from these studies or what has already been shown by others.

We have re-written this part of the discussion, clearly separating our own data from previous findings.

14) Reviewer #3, Point 7:

Skips have been proposed to have an important role in the production of Schwann cells

following injury and in cancer. The role of these cells should be discussed in the context of this study.

We feel that there has been a confusion in the field because SKPs were originally thought to be neural crest derivatives and later found that, in the trunk, they actually originate from the mesoderm, unlike neural crest-derived peripheral glia. In the revised Discussion on p.19, we now cite the corresponding literature (Biernaskie et al., 2009) and also discuss the possibility that activated glia might interact with other stem and progenitor cells types, such as SKPs, given that Schwann cell precursors purified from neonatal sciatic nerve can stimulate self-renewal of mesodermal SKPs (Johnston et al., 2016).

15) Reviewer #4, General Points:

The manuscript by Parfejevs et al is very carefully executed, analyzed, presented and written. The main conclusion is timely and exciting. I find it a bit surprising that Schwann cells don't contribute to melanocytes during healing, but the results seem robust. Perhaps it is the model they use, and contribution is greater in cases of hyperpigmentation associated with healed wounds?

We are grateful for this reviewer's very positive comments. We were also surprised to see virtually no melanocytes developing from injury-activated glia, in particular given the intriguing literature by Adameyko and colleagues on the developmental potential of peripheral glia. This literature, cited in our study, actually prompted us to carefully monitor melanocytic markers in genetically traced cells in the wound (see also our revised Supplementary Figure 2, our Point 8) above). Furthermore, we used *Tyr-CreERT2*-mediated melanocyte-lineage tracing to confirm the data. Indeed, peripheral glia might realize their melanocytic potential in other conditions (see also Discussion p.17).

16) Reviewer #4, Point 1: see also our Point 5) above

It is unclear what makes some nerve bundles active and other inactive. Is there a correlation between lesioned axon bundles and active bundles? It would be good if the authors could address this issue experimentally, or at least speculate.

As also explained in our reply to the same issue raised by Reviewer #2, dissection and isolation of peripheral nerves was sufficient to trigger glia de-differentiation and their dissemination from nerves, closely mimicking the processes we observed upon skin wounding in vivo. As shown in new Figure 7b, when placing acutely dissected peripheral nerves in culture, genetically traced glial cells started to express p75 and c-Jun and

emigrated from the nerves, thus displaying the hallmarks of injury-induced glia activation in skin wounds *in vivo*. This was also the case when placing isolated peripheral nerves directly on culture dishes rather than on fibroblasts (data not shown). As now discussed on p16., we conclude that presence of other cell types (such as immune cells and other dermal cells) is apparently not an absolute prerequisite for glial cell activation; rather, direct physical trauma seems to be sufficient to trigger this process, although the exact molecular mechanism underlying glial cell activation remains to be determined.

17) Reviewer #4, Point 2:

In general, the genetic animal models are very superficially explained in the main result. I think that most readers would want to understand what models were used and how they were used. The authors need to include the full genotype of each mouse model the first time it is used and explain how it will be abbreviated in the rest of the manuscript. For example, it is unclear how Sox10 is conditionally deleted using floxed alleles and the same time used for lineage tracing. Information on tamoxifen concentrations, administration regime and time at analysis is not always explained.

We now tried to clarify these issues as much as possible in the Figures and in the text, including the Methods section.

18) Reviewer #4, Point 3:

Similar issues relates to markers used, such as CD31, CD45 and more. These are never even mentioned in the main text. All information on markers used and what cell types they are believed to stain needs to be included in the result section. Similarly, please explain how (what) hyperproliferation was quantified.

As for the previous point by this reviewer, we now tried to clarify these issues as much as possible in the Figures and in the text. In particular, we added further labelling in the figures to clarify genotypes and markers used, to indicate wounded vs non-wounded samples, structures shown, and time points of analysis, etc. (see also our Point 9) above).

19) Reviewer #4, Point 4:

It would have been nice to see an ablation experiment instead of Sox10 cKO strategy for establishing the role of these cells for wound healing. Is it possible that depletion of Sox10 changes the fate of the cells so they are still there, just with a different phenotype? There are protocols for topical administration of hydroxytamoxifen, so a local ablation strategy should be feasible. It would be nice if the authors could address this either experimentally or by

clarifying what is actually believed to be happening with the Schwann cells when Sox10 is deleted.

We have originally considered local ablation of *Sox10* or also local depletion of glia by means of lineage-specific expression of diphtheria toxin. However, local genetic manipulation turned out to yield much lower recombination efficiencies than achieved upon systemic tamoxifen treatment; moreover, locally induced genetic recombination involves topical application of 4OH-tamoxifen, which induced considerable adverse skin inflammation in our experimental settings (in agreements with the supplier's product description). As this could obviously affect both read-out of the experiments and interpretation of the data, one has to refrain in our opinion from any inflammation-inducing agents in a study aiming to address the role of glia in wound healing. Diphtheria-induced systemic depletion of all peripheral glia would likely result in a much more drastic phenotype than what we obtain in *Sox10* cko mice, which would make interpretation of the data more difficult if not impossible.

Therefore, we have used conditional knock out of *Sox10* to deplete activated glia, knowing from previous work by others and us that this affects survival of neural crest-derived progenitor cells and Schwann cell precursors. In contrast, it has been shown that it takes a while until fully differentiated glia lose *Sox10* and eventually die upon lineage-specific genetic ablation of *Sox10* (Bremer et al., 2011). However, since the main concern of the reviewer was that *Sox10* loss might not result in loss of injury-activated peripheral glia, as we proposed based on the available literature, but rather leads to cell fate changes, we have now performed lineage fate mapping of control and *Sox10* conditional knock out (cko) cells upon wounding. As shown in novel Supplementary Figure 3, *Sox10* cko cells do not acquire alternative fates but are indeed lost both in nerve bundles and in the wound bed. Therefore, our genetic system allows assessing the function of activated peripheral glia during wound healing by *Sox10* cko-mediated depletion of this cell population.

20) Reviewer #4, Point 5:

Ref 54, Johnston et al., although examined tip regeneration has used a similar strategy. It would be good if the present study could be put into this context to a greater extent in the discussion.

We are not exactly sure to what further extent this paper should be discussed in addition to our points already made, but in trying to follow this reviewer's suggestion we somewhat extended our Discussion of the recent paper by Johnston et al. (p.19).

We would like to again express our thanks for your very constructive and supportive comments. We hope that with the new data and further improvements of the manuscript, our study will now be acceptable for publication in *Nature Communications*.

REVIEWERS' COMMENTS:

Reviewer #1:

{ED: This referee did not have formal comments to the authors as s/he found the revised paper is satisfactory and endorses publication.}

Reviewer #2 (Remarks to the Author):

The authors have very adequately answered the comments of all referees, and this manuscript would now be suitable for publication in Nature Communications in the opinion of this referee.

Reviewer #3 (Remarks to the Author):

I have read the revised manuscript and find that the authors have satisfactorily addressed all of my comments and that the manuscript is now suitable for publication.

Reviewer #4 (Remarks to the Author):

The authors have fully addressed all my concerns. This is a very exciting study that I hope is published soon.